# Efficient Training of Multi-task Neural Solver for Combinatorial Optimization

**Chenguang Wang**                                                    *chenguangwang@link.cuhk.edu.cn*
*School of Data Science, The Chinese University of Hong Kong, Shenzhen*
*Shenzhen Research Institute of Big Data*

**Zhang-Hua Fu**[†]                                                    *fuzhanghua@cuhk.edu.cn*
*School of Science and Engineering, The Chinese University of Hong Kong, Shenzhen*
*Shenzhen Institute of Artificial Intelligence and Robotics for Society*

**Pinyan Lu**                                                        *lu.pinyan@mail.shufe.edu.cn*
*Shanghai University of Finance and Economics*

**Tianshu Yu**[†]                                                    *yutianshu@cuhk.edu.cn*
*School of Data Science, The Chinese University of Hong Kong, Shenzhen*

**Reviewed on OpenReview:** *https://openreview.net/forum?id=HJbcwRbMQQ*

## Abstract

Efficiently training a multi-task neural solver for various combinatorial optimization problems (COPs) has been less studied so far. Naive application of conventional multi-task learning approaches often falls short in delivering a high-quality, unified neural solver. This deficiency primarily stems from the significant computational demands and a lack of adequate consideration for the complexities inherent in COPs. In this paper, we propose a general and efficient training paradigm to deliver a unified combinatorial multi-task neural solver. To this end, we resort to the theoretical loss decomposition for multiple tasks under an encoder-decoder framework, which enables more efficient training via proper bandit task-sampling algorithms through an intra-task influence matrix. By employing theoretically grounded approximations, our method significantly enhances overall performance, regardless of whether it is within constrained training budgets, across equivalent training epochs, or in terms of generalization capabilities, when compared to conventional training schedules. On the real-world datasets of TSPLib and CVRPLib, our method also achieved the best results compared to single task learning and multi-task learning approaches. Additionally, the influence matrix provides empirical evidence supporting common practices in the field of learning to optimize, further substantiating the effectiveness of our approach. Our code is open-sourced and available at `https://github.com/LOGO-CUHKSZ/MTL-COP`.

## 1 Introduction

Combinatorial optimization problems (COPs) (Korte et al., 2011) are fundamental in various fields, including logistics (Cattaruzza et al., 2017; Bao et al., 2018), finance (Tatsumura et al., 2023), telecommunications (Resende & Pardalos, 2008), and computer science, where they are essential for making optimal decisions over discrete structures. Traditional methods for solving COPs, such as exact algorithms (Wolsey, 2020), approximation methods (Vazirani, 2001) and heuristics (Boussaïd et al., 2013), often face significant challenges due to their computational complexity and inefficiency, especially in large-scale and dynamic scenarios. The emergence of deep learning has introduced a transformative approach by leveraging powerful modeling capabilities to learn from data and generalize solutions across similar instances. Building on this idea, many

---

[†]Corresponding author

studies have attempted to use machine learning-based methods, such as supervised learning (Vinyals et al., 2015; Fu et al., 2021), reinforcement learning (Bello et al., 2017; Kool et al., 2019; Kwon et al., 2020; Lu et al., 2020; Wu et al., 2021b), or unsupervised learning (Hibat-Allah et al., 2021; Schuetz et al., 2022; Sanokowski et al., 2023; 2024), to solve COPs by proposing neural solvers. However, these approaches typically focus on a single type of combinatorial optimization problem, or even a specific scale of that problem, significantly limiting the feasibility of neural solvers in practical applications. Therefore, training a neural solver capable of handling different types and sizes of COPs simultaneously is an urgent issue that needs to be addressed. Recently, some research has begun to address this issue, proposing the use of a unified neural solver to tackle different problems such as Vehicle Routing Problems (VRPs) (Liu et al., 2024; Lin et al., 2024; Berto et al., 2024; Zhou et al., 2024) and graph-based COPs (Boisvert et al., 2024). However, these solutions are often restricted to a specific category of problems. For instance, they may leverage specialist knowledge to construct models based on the combinatorial characteristics of various VRPs (such as VRPTW, OVRP, VRPB) or focus solely on graph-based COPs. As a result, they fail to develop a neural solver that can simultaneously solve multiple types of COPs.

Although a generic neural solver for multiple COPs is appealing, training such a neural solver can be prohibitively expensive, especially in the era of large models and this problem is less studied in the literature. To relieve the training burden and better balance the resource allocation, in this paper, we propose a novel training paradigm from a multi-task learning (MTL) perspective, which can efficiently train a multi-task combinatorial neural solver to handle different types of COPs under limited training budgets.

To this end, we treat each COP with a specific problem scale as a task and manage to deliver a generic solver handling a set of tasks simultaneously. Different from a standard joint training in MTL, we employ multi-armed bandits (MAB) algorithms to select/sample one task in each training round, hence avoiding the complex balancing of losses from multiple tasks. To better guide the MTL training, we employ a reasonable reward design derived from the theoretical loss decomposition to derive the rewards with theoretical guarantees in MAB algorithms for the widely adopted encoder-decoder architecture. This loss decomposition also brings about an influence matrix revealing the mutual impacts between tasks, which provides rich evidence to explain some common practices in the scope of COPs.

To emphasize, our method is the first-of-its-kind to consider training a generic neural solver for different kinds of COPs. This greatly differs from existing works focusing on either solution construction (Vinyals et al., 2015; Bello et al., 2017; Kool et al., 2019; Kwon et al., 2020) or heuristic improvement (Lu et al., 2020; Wu et al., 2021b; Agostinelli et al., 2021; Fu et al., 2021; Kool et al., 2022). Some recent works seek to generalize neural solvers to different scales (Hou et al.; Li et al., 2021; Cheng et al., 2023; Wang et al., 2024) or varying distributions (Wang et al., 2021; Bi et al., 2022; Geisler et al., 2022), but with no ability to handle multiple types of COPs simultaneously. Compared to methods aimed at building a universal neural solver (Liu et al., 2024; Lin et al., 2024; Berto et al., 2024; Zhou et al., 2024; Boisvert et al., 2024), our focus is a generalized training framework that does not require expert knowledge during the training phase, nor does it necessitate specifying the type of problems. This framework is suitable for any neural solver and is adaptable to a broad range of COPs

Experiments are conducted for 12 tasks: Four types of COPs, the Travelling Salesman Problem (TSP), the Capacitated Vehicle Routing Problem (CVRP), the Orienteering Problem (OP), and the Knapsack Problem (KP), and each of them with three problem scales. We compare our approach with single-task training (STL), extensive MTL baselines (Mao et al., 2021; Yu et al., 2020; Navon et al., 2022; Kendall et al., 2018; Liu et al., 2021a;b) and the SOTA task grouping method, TAG (Fifty et al., 2021) under the cases of the same training budgets and same training epochs. Compared with STL, our approach needs no prior knowledge about tasks and can automatically focus on harder tasks to maximally utilize the training budget. Additionally, when comparing with STL under the same training epoch, our approach not only enjoys the cheaper training cost which is strictly smaller than that of the most expensive task, but also shows the generalization ability by providing a universal model to cover different types of COPs. Compared with the MTL methods, our method only picks the most impacting task to train at each time. This mechanism improves the training efficiency without explicitly balancing the losses. Furthermore, we also compare our approach with STL and MTL methods on real-world datasets, TSPLib and CVRPLib, achieving the best experimental performance.

In summary, our contributions can be concluded as follows: **(1)** We propose a novel framework for efficiently training a combinatorial neural solver for multiple COPs via MTL, which achieves prominent performance against standard training paradigms with limited training resources and can further advise efficient training of other large models; **(2)** We study the intrinsic loss decomposition for the well-adopted encoder-decoder architecture and propose a theoretically guaranteed approximation for it, leading to the influence matrix reflecting the inherent task relations and reasonable reward guiding the update of MAB algorithms; **(3)** We verify several empirical observations for neural solvers from previous works (Kool et al., 2019; Joshi et al., 2021) by the influence matrix, demonstrating the validity and reasonableness of our approach; **(4)** We compare our method to a wide range of baselines on synthetic and real-world datasets, demonstrating that it outperforms STL and MTL approaches in efficiency and results.

## 2 Related Work

**Neural solvers for COPs** Pointer Networks (Vinyals et al., 2015) pioneered the application of deep neural networks for solving combinatorial optimization problems. Subsequently, numerous neural solvers have been developed to address various COPs, such as routing problems (Bello et al., 2017; Kool et al., 2019; Lu et al., 2020; Wu et al., 2021b;b), knapsack problem (Bello et al., 2017; Kwon et al., 2020), job shop scheduling problem (Zhang et al., 2020), and others. There are two prevalent approaches to constructing neural solvers: solution construction (Vinyals et al., 2015; Bello et al., 2017; Kool et al., 2019; Kwon et al., 2020), which sequentially constructs a feasible solution, and heuristic improvement (Lu et al., 2020; Wu et al., 2021b; Agostinelli et al., 2021; Fu et al., 2021; Kool et al., 2022), which provides meaningful information to guide downstream classical heuristic methods. In addition to developing novel techniques, several works (Wang et al., 2021; Geisler et al., 2022; Bi et al., 2022; Wang et al., 2024) have been proposed to address generalization issues inherent in COPs. For a comprehensive review of the existing challenges in this area, we refer to the survey Bengio et al. (2020).

**Multi-task learning** Multi-Task Learning (MTL) aims to enhance the performance of multiple tasks by jointly training a single model to extract shared knowledge among them. Numerous works have emerged to address MTL from various perspectives, such as exploring the balance on the losses from different tasks (Mao et al., 2021; Yu et al., 2020; Navon et al., 2022; Kendall et al., 2018; Liu et al., 2021a;b) designing module-sharing mechanisms (Misra et al., 2016; Sun et al., 2020; Hu & Singh, 2021), improving MTL through multi-objective optimization (Sener & Koltun, 2018; Lin et al., 2019; Momma et al., 2022), and meta-learning (Song et al., 2022). To optimize MTL efficiency and mitigate the impact of negative transfer, some research focuses on task-grouping (Kumar & III, 2012; Zamir et al., 2018; Standley et al., 2020; Fifty et al., 2021), with the goal of identifying task relationships and learning within groups to alleviate negative transfer effects in conflicting tasks. On the application level, MTL has been extensively employed in various domains, including natural language processing (Collobert & Weston, 2008; Luong et al., 2016), computer vision (Zamir et al., 2018; Seong et al., 2019), bioinformatics (Xu et al., 2017), and many others.

**Foundation models for COPs** Recently, research has increasingly focused on foundation models for combinatorial optimization problems. For instance, studies such as Liu et al. (2024); Lin et al. (2024); Berto et al. (2024); Zhou et al. (2024) tackle cross-problem generalization in vehicle routing problems (VRPs) using unified neural solvers that leverage attribute composition. However, these approaches rely on the combinatorial nature of VRP structures and necessitate specific expert knowledge for problem decomposition, limiting their applicability to other types of COPs. Meanwhile, Boisvert et al. (2024) proposes a generic representation for COPs via graph-based approaches, facilitating efficient learning with a novel graph neural network architecture. Additionally, Drakulic et al. (2024) presents a generalist model that efficiently addresses multiple COPs using a mixed-attention backbone and multi-type transformer architecture, achieving task-specific adaptability through lightweight adapters and showcasing strong transfer learning capabilities. Furthermore, Jiang et al. (2024) introduces a model that solves diverse combinatorial optimization problems using natural language-formulated instances and a large language model for embedding, enhanced by an encoder-decoder architecture and trained with conflict gradients erasing reinforcement learning, thereby improving performance across tasks.

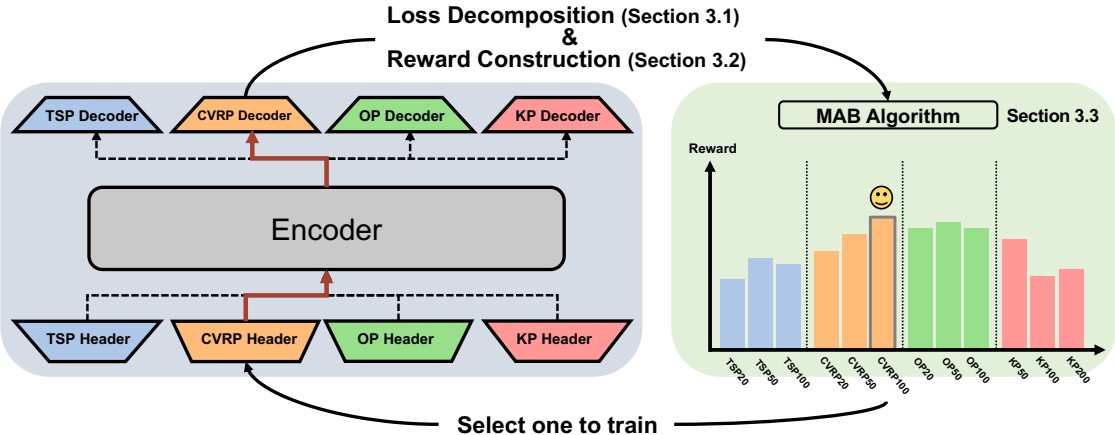

Figure 1: Pipeline of MAB for Solving COPs in view of MTL. We consider four types of COPs: TSP, CVRP, OP and KP, each with a corresponding header and decoder. The encoder, which is common to all COPs, is also included. For each time step, we utilize the MAB algorithm to select a specific task for training, such as CVRP-100 depicted in the figure. We then obtain the loss for the selected task, perform loss decomposition as detailed in Section 3.1, and construct a reward using the methodology outlined in Section 3.2. Finally, we utilize the reward to update the MAB algorithm.

**Multi-armed bandits** Multi-armed bandit (MAB) is a classical problem in decision theory and machine learning that addresses the exploration-exploitation trade-off. Several algorithms and strategies have been suggested to solve the MAB problem, such as the $\epsilon$-greedy, Upper Confidence Bound (UCB) family of algorithms (Lai et al., 1985; Auer et al., 2002), the Exp3 family (Littlestone & Warmuth, 1994; Auer et al., 1995; Gur et al., 2014), and the Thompson sampling (Thompson, 1933; Agrawal & Goyal, 2012; Chapelle & Li, 2011). These methods differ in their balance of exploration and exploitation, and their resilience under distinct types of uncertainty. The MAB has been extensively studied in both theoretical and practical contexts, and comprehensive details can be found in Slivkins et al. (2019); Lattimore & Szepesvári (2020).

## 3 Method

We consider $K$ types of COPs, denoted as $T^i$ $(i = 1, 2, ..., K)$, with $n_i$ different problem scales for each COP. Thus, the overall task set is $\mathcal{T} = \bigcup_{i=1}^{K} T^i := \{T_j^i | j = 1, 2, ..., n_i, i = 1, 2, ..., K\}$. For each type of COP $T^i$, we consider a neural solver $S_{\Theta^i}(\mathcal{I}_j^i) : T_j^i, \mathcal{Y}_j^i$, where $\Theta^i$ are the parameters for COP $T^i$, $\mathcal{I}_j^i$ and $\mathcal{Y}_j^i$ are the input instance the output space for COP $T^i$ with the problem scale of $n_j$ (termed as task $T_j^i$ in the sequel). The parameter vector $\Theta^i = (\theta^{\text{share}}, \theta^i)$ contains the shared and task-specific parameters for the COP $T^i$, and the complete set of parameters is denoted by $\Theta = \bigcup_{i=1}^{K} \Theta^i$. This parameter notation corresponds to the commonly used Encoder-Decoder framework [1] in multi-task learning in Figure 1, where $\theta^{\text{share}}$ represents the encoder - shared across all tasks, and $\theta^i$ represents the decoder - task-specific for each task. Given the task loss functions $L_j^i(\Theta^i)$ for COP $T^i$ with the problem scale of $n_j$, we investigate the widely used objective function:

$$\min_{\Theta} L(\Theta) = \sum_{i=1}^{K} \sum_{j=1}^{n_i} L_j^i(\Theta^i). \tag{1}$$

We propose a general Multi-task Learning (MTL) framework based on Multi-Armed Bandits (MAB) to dynamically select tasks during training rounds and a reasonable reward is constructed to guide the selection process. In particular, our approach establishes a comprehensive task relation by the obtained influence

---

[1] According to the Encoder-Decoder framework, encoder commonly refers to shared models, whereas decoder concerns task-specific modules. In this study, the decoder component comprises two modules: "Header" and "Decoder" as illustrated in Figure 1.

matrix, which has the potential to empirically validate several common deep learning practices while solving COPs.

**Overview** We aim to solve equation 1 using the MAB approach. Given the set of tasks $\mathcal{T} = \{T_j^i | j = 1, 2, ..., n_i, i = 1, 2, ..., K\}$, we select an arm (i.e., task being trained) $a_t \in \mathcal{T}$ following an MAB algorithm, which yields a random reward signal $r_t$ that reflects the effect of the selection. The approximated expected reward is updated based on the rewards. Essentially, our proposed method is applicable to any MAB algorithm. The general framework of MAB algorithm for solving COPs within the context of MTL is outlined in Figure 1.

## 3.1 Loss Decomposition

In the framework of solving COPs in view of MTL described in Figure 1, it is crucial to design a reasonable reward for MAB algorithm to guide its update. In this part, we analytically drive a reasonable reward by decomposing the loss function for the Encoder-Decoder framework . Note $\Theta = \bigcup_{i=1}^{K} \Theta^i = \{\theta^{\text{share}}\} \bigcup \{\theta_i, i = 1, 2, ..., K\}$ are all trainable parameters.

We suppose that a meaningful reward should satisfy the following two properties: **(1)** It can benefit our objective and reveal the intrinsic training signal; **(2)** When a task is selected, there is always positive effect on it in expectation.

The decrease on loss function is a reasonable choice and previous work has used it to measure the task relationship (Fifty et al., 2021). However, such measurement is invalid in our context because there are no significant differences among tasks (see Appendix A.8) . What's more, the computational cost of the *"lookahead loss"* in TAG (Fifty et al., 2021) is considerably expensive when frequent reward signals are needed. We instead propose a more fundamental way based on gradients to measure the impacts of training one task upon the others.

To simplify the analysis, in Proposition 1 we assume the standard gradient descent (GD) is used to optimize equation 1 by training one task at each step $t$, and then derive the loss decomposition under the encoder-decoder framework. Any other optimization methods, e.g., Adam (Kingma & Ba, 2015), can also be used here with small modifications. Detailed proofs for GD and Adam are in Appendix A.2.

**Proposition 1 (Loss decomposition for GD)** *Using encoder-decoder framework with parameters* $\Theta = \bigcup_{i=1}^{K} \Theta^i = \{\theta^{share}\} \bigcup \{\theta_i, i = 1, 2, ..., K\}$ *and updating parameters with standard gradient descent:* $\Theta(t+1) = \Theta(t) - \eta_t \nabla L(\Theta(t))$, *where* $\eta_t$ *is the step size. Then the difference of the loss of task* $T_j^i$ *from training step* $t_1$ *to* $t_2$: $\Delta L_j^i(t_1, t_2) = L_j^i(\Theta^i(t_2)) - L_j^i(\Theta^i(t_1))$ *can be decomposed to:*

$$
\begin{aligned}
&\Delta L_j^i(t_1, t_2) \\
&= -\Big( \underbrace{\sum_{t=t_1}^{t_2-1} \eta_t \mathbb{1}(a_t = T_j^i) \nabla^T L_j^i(\Psi^i(t)) \nabla L_j^i(\Theta^i(t))}_{(a)\ \textit{effects of training task } T_j^i:\ e_j^i(t_1,t_2)} + \underbrace{\sum_{\substack{q=1 \\ q \neq j}}^{n_i} \sum_{t=t_1}^{t_2-1} \eta_t \mathbb{1}(a_t = T_q^i) \nabla^T L_j^i(\Psi^i(t)) \nabla L_q^i(\Theta^i(t))}_{(b)\ \textit{effects of training task } \{T_q^i, q \neq j\}:\ \{e_q^i(t_1,t_2), q \neq j\}} \\
&+ \underbrace{\sum_{\substack{p=1 \\ p \neq i}}^{K} \sum_{q=1}^{n_p} \sum_{t=t_1}^{t_2-1} \eta_t \mathbb{1}(a_t = T_q^p) \nabla^T_{\theta^{share}} L_j^i(\Psi^i(t)) \nabla_{\theta^{share}} L_q^p(\Theta^p(t)) \Big)}_{(c)\ \textit{effects of training task } \{T_q^p, p \neq i\}:\{e_q^p(t_1,t_2), q=1,2,...,n_p, p \neq i\}},
\end{aligned}
\tag{2}
$$

*where* $\nabla L(\Theta)$ *means taking gradient w.r.t.* $\Theta$ *and* $\nabla L_\theta(\Theta)$ *means taking gradient w.r.t.* $\theta \subseteq \Theta$, $\Psi^i(t)$ *is some vector lying between* $\Theta^i(t)$ *and* $\Theta^i(t)$ *and* $\mathbb{1}(a_t = T_j^i)$ *is the indicator function.*

**Proof 1 (Proof of proposition 1:)** *Observe that*

$$\Delta L_j^i(t_1, t_2) = \sum_{t=t_1}^{t_2-1} \left[ \Delta L_j^i(t, t+1) \left( \sum_{p=1}^{K} \sum_{q=1}^{n_p} \mathbb{1}(a_t = T_q^p) \right) \right] = \sum_{t=t_1}^{t_2-1} \mathbb{1}(a_t = T_j^i) \Delta L_j^i(t, t+1)$$

$$+ \sum_{t=t_1}^{t_2-1} \sum_{\substack{q=1 \\ q \neq j}}^{n_i} \mathbb{1}(a_t = T_q^i) \Delta L_j^i(t, t+1) + \sum_{t=t_1}^{t_2-1} \sum_{\substack{p=1 \\ p \neq i}}^{K} \sum_{q=1}^{n_p} \mathbb{1}(a_t = T_q^p) \Delta L_j^i(t, t+1). \tag{3}$$

where $\mathbb{1}(a_t = T_j^i)$ is the indicator function which is introduced here because we only select one task at each time step, taking 1 if selecting task $T_j^i$ at time step $t$, 0 otherwise. Then based on mean value theorem, we take the first order Taylor expansion for

$$\mathbb{1}(a_t = T_q^p) \Delta L_j^i(t, t+1) = \begin{cases} - \eta_t \nabla^T L_j^i(\Psi^i(t)) \nabla L_q^i(\Theta^i(t)) & \text{if } p = i \\ - \eta_T \nabla_{\theta^{share}}^T L_j^i(\Psi^i(t)) \nabla_{\theta^{share}}^T L_q^p(\Theta^p(t)) & \text{Otherwise} \end{cases} \tag{4}$$

where $\Psi^i(t)$ is some vector lying between $\Theta^i(t)$ and $\Theta^i(t+1)$. Suppose task $T_j^i$ is selected for $c_j^i$ times between time step $t_1$ and $t_2$. After plugging equation 4 in equation 3, we obtain

$$L_j^i(\Theta^i(t_2)) - L_j^i(\Theta^i(t_1)) = \underbrace{-(\nabla^T L_j^i(\Psi^i(t_1)) \sum_{t=t_1}^{t_2} \mathbb{1}(a_t = T_j^i) \eta_t \nabla L_j^i(\Theta^i(t)) +}_{(a) \text{ effects of training task } T_j^i: \ e_j^i(t_1,t_2)}$$

$$\underbrace{\nabla^T L_j^i(\Psi^i(t_1)) \sum_{\substack{q=1 \\ q \neq j}}^{n_i} \sum_{t=t_1}^{t_2} \mathbb{1}(a_t = T_q^i) \eta_t \nabla L_q^i(\Theta^i(t))}_{(b) \text{ effects of training task } \{T_q^i, q \neq j\}: \ \{e_q^i((t_1,t_2)), q \neq j\}} + \underbrace{\nabla_{\theta^{share}}^T L_j^i(\Psi^i(t_1)) \sum_{\substack{p=1 \\ p \neq i}}^{K} \sum_{q=1}^{n_p} \sum_{t=t_1}^{t_2} \mathbb{1}(a_t = T_q^p) \eta_t \nabla_{\theta^{share}}^T L_q^p(\Theta^p(t)))}_{(c) \text{ effects of training task } \{T_q^p, p \neq i\}: \{e_q^p(t_1,t_2), q=1,2,\ldots,n_p, p \neq i\}}. \tag{5}$$

The idea behind equation 2 means the improvement on the loss for task $T_j^i$ from $t_1$ to $t_2$ can be decomposed into three parts: (**a**) effects of training $T_j^i$ itself w.r.t. $\Theta^i$; (**b**) effects of training same kind of COP $\{T_q^i, q \neq j\}$ w.r.t. $\Theta^i$; and (**c**) effects of training other COPs $\{T^p, p \neq i\}$ w.r.t. $\theta^{share}$. Indeed, we quantify the impact of different tasks on $T_j^i$ through this decomposition, which provides the intrinsic training signals for designing reasonable rewards.

## 3.2 Reward Design and Influence Matrix Construction

In this part, we design the reward and construct the intra-task relations based on the loss decomposition introduced in Section 3.1. Though equation 2 reveals the signal during training, the inner products of gradients from different tasks can significantly differ at scale (see Appendix A.8). This may greatly mislead the bandit's update since improvements may come from large gradient values even when they are almost orthogonal. To address this, we propose to use cosine metric to measure the influence between task pairs. Formally, for task $T_j^i$ from $t_1$ to $t_2$, the influence from training the same type of COP $T_q^i$ to $T_j^i$ is:

$$m_j^i(T_q^i; t_1, t_2) = \sum_{t=t_1}^{t_2-1} \frac{\mathbb{1}(a_t = T_q^i)}{\sum_{t=t_1}^{t_2-1} \mathbb{1}(a_t = T_q^i)} \frac{\nabla^T L_j^i(\Psi^i(t)) \nabla L_q^i(\Theta^i(t))}{||\nabla^T L_j^i(\Psi^i(t))|| \cdot ||\nabla L_q^i(\Theta^i(t))||}, \tag{6}$$

and the influence from training other types of COPs $T_q^p$ to $T_j^i$ is:

$$m_j^i(T_q^p; t_1, t_2) = \sum_{t=t_1}^{t_2-1} \frac{\mathbb{1}(a_t = T_q^p)}{\sum_{t=t_1}^{t_2-1} \mathbb{1}(a_t = T_q^p)} \frac{\nabla_{\theta^{share}}^T L_j^i(\Psi^i(t)) \nabla_{\theta^{share}} L_q^p(\Theta^p(t))}{||\nabla_{\theta^{share}}^T L_j^i(\Psi^i(t))|| \cdot ||\nabla_{\theta^{share}} L_q^p(\Theta^p(t))||}. \tag{7}$$

Given equation 6 and equation 7, we denote the influence vector to $T_j^i$ as:

$$\mathbf{m}_j^i(t_1, t_2) = (..., m_j^i(T_1^i; t_1, t_2), ..., \underbrace{\overbrace{m_j^i(T_j^i; t_1, t_2)}^{\text{influence from itself}}, ..., m_j^i(T_{n_i}^i; t_1, t_2)}_{\text{influence from the same kind of COP}}, \underbrace{..., m_j^i(T_q^p; t_1, t_2), ...}_{\text{influence from other kinds of COPs}} )^T \tag{8}$$

Based on equation 8, an influence matrix $M(t_1, t_2) = (..., \mathbf{m}_j^i(t_1, t_2), ...)^T \in \mathbb{R}^{(\sum_{k=1}^{K} n_k) \times (\sum_{k=1}^{K} n_k)}$ can be constructed to reveal the relationship between tasks from time step $t_1$ to $t_2$. There are several properties about influence matrix $M(t_1, t_2)$: **(1)** $M(t_1, t_2)$ has blocks $M^i(t_1, t_2) \in \mathbb{R}^{n_i}$ in the diagonal position which is the sub-influence matrix of a same kind of COP with different problem scales; **(2)** $M(t_1, t_2)$ is asymmetry which is consistent with the general understanding in multi-task learning; **(3)** The row-sum of $M(t_1, t_2)$ are the total influences obtained *from* all tasks *to* one task; **(4)** The column-sum of $M(t_1, t_2)$ are the total influences *from* one task *to* all tasks.

According to the implication of the elements in $M(t_1, t_2)$, the column-sum of $M(t_1, t_2)$:

$$\mathbf{r}(t_1, t_2) = \mathbf{1}^T \cdot M(t_1, t_2) \in \mathbb{R}^{1 \times \sum_{k=1}^{K} n_k} \tag{9}$$

provides a meaningful reward signal for selecting tasks ,which we can use to update the bandit algorithm. Moreover, we denote the update frequency of computing the influence matrix as $\Delta T$ and the overall training time is $n\Delta T$, then an average influence matrix $W$ can be constructed based on influence matrices $\{M(k\Delta T, (k+1)\Delta T), k = 0, 2, ..., n-1\}$ collected during the training process:

$$W = \frac{1}{n\Delta T} \sum_{k=1}^{n-1} M\left(k\Delta T, (k+1)\Delta T\right), \tag{10}$$

revealing the overall task relations across the training process.

### 3.3 Theoretical Analysis and Implementation

When computing the bandit rewards, an issue arises with approximating $\nabla^T L_j^i(\Psi^i(t))$ in Eqs. equation 6 and equation 7. A practical and justifiable approach involves using the approximation $\nabla^T L_j^i(\Psi^i(t)) \approx \nabla^T L_j^i(\Theta^i(t))$ based on a Taylor expansion, as the parameter changes during neural network updates are typically minimal. Nonetheless, obtaining this gradient information is challenging because only one task is selected per training slot. To address this, we adopt the approximation

$$\nabla^T L_j^i(\Theta^i(\tau_{i,j}(t))) \approx \nabla^T L_j^i(\Theta^i(t))$$

where $\tau_{i,j}(t)$ represents the most recent training slot for $T_j^i$. This method is straightforward to implement and offers the following theoretical guarantee.

**Theorem 1** *For each task, suppose there exists $M > 0$ such that $||\nabla^2 L_j^i(\Theta^i)|| \leq M$. Moreover, we suppose the gradients have equal norm across the training period. For any task $T_q^p$, we denote $\hat{m}_j^i(T_q^p; t_1, t_2)$ and $\tilde{m}_j^i(T_q^p; t_1, t_2)$ as the approximations of using $\nabla^T L_j^i(\Theta^i(\tau_{i,j}(t)))$ and $\nabla^T L_j^i(\Theta^i(t))$ in $m_j^i(T_q^p; t_1, t_2)$, then*

$$|\hat{m}_j^i(T_q^p; t_1, t_2) - \tilde{m}_j^i(T_q^p; t_1, t_2)| \leq \frac{M \eta^{max}(t_1, t_2)}{c_q^p(t_1, t_2)} \sum_{t=t_1}^{t_2} \mathbb{1}(a_t = T_q^p)(t - \tau_{i,j}(t))$$

*where $\eta^{max}(t_1, t_2)$ and $c_q^p(t_1, t_2)$ are the maximal learning rate and selection times of $T_q^p$ from $t_1$ to $t_2$.*

In practice, this upper bound is reasonably negligible due to the small value of $M$ (Sagun et al., 2017; Ghorbani et al., 2019; Yao et al., 2020; Li et al., 2020) and $\frac{\eta^{max}(t_1, t_2)}{c_q^p(t_1, t_2)} \sum_{t=t_1}^{t_2} \mathbb{1}(a_t = T_q^p)(t - \tau_{i,j}(t))$, indicating a good approximation during the implementation. We leave the proofs and discussions in Appendix A.3

### 3.4 Computation Complexity of MTL Methods

In this part, we will make a detailed comparison on the computation complexity between our method and other typical MTL methods. We first define some notations for the time complexity: $N$, $D_i$, $F_i$ and $B_i$ where $D_i, F_i$ and $B_i$ are the dimension of parameters, the computation cost of feed-forward and backward for task $i$,

Table 1: Computation complexity of different MTL methods for one training time slot. "Basic" measures the computation for the feed-forward and backward process, "Extra" measures the extra computations used for guiding MTL, and "All" is the sum of them.

| | Naive-MTL | Bandit-MTL | PCGrad | Nash-MTL | UW | IMTL | CAGrad | Ours |
|---|---|---|---|---|---|---|---|---|
| Basic | $\mathcal{O}(N(F+B))$ | $\mathcal{O}(N(F+B))$ | $\mathcal{O}(N(F+B))$ | $\mathcal{O}(N(F+B))$ | $\mathcal{O}(N(F+B))$ | $\mathcal{O}(N(F+B))$ | $\mathcal{O}(N(F+B))$ | $\mathcal{O}(\mathbf{F}+\mathbf{B})$ |
| Extra | 0 | $\mathcal{O}(\mathbf{N})$ | $\mathcal{O}(N^2D)$ | $\mathcal{O}(N^2D)$ | $\mathcal{O}(1)$ | $\mathcal{O}(N^2D+N^3)$ | - | $\mathcal{O}(ND)$ |
| All | $\mathcal{O}(N(F+B))$ | $\mathcal{O}(N(F+B+1))$ | $\mathcal{O}(N(F+B+ND))$ | $\mathcal{O}(N(F+B+ND))$ | $\mathcal{O}(N(F+B))$ | $\mathcal{O}(N(F+B+N^2+ND))$ | - | $\mathcal{O}(\mathbf{F}+\mathbf{B}+\mathbf{ND})$ |
| GPU Hours | 1.00 | 1.04 | 6.02 | 5.87 | 1.00 | 5.61 | 5.24 | 0.07 |

and we denote $D = \max\{D_i, i = 1, 2, ..., N\}, F = \max\{F_i, i = 1, 2, ..., N\}, B = \max\{B_i, i = 1, 2, ..., N\}$. We analyze the computation complexity for Bandit-MTL, PCGrad, Nash-MTL, Uncertainty-Weighting (UW) and our method, results are shown in Table1, where "Basic" measures the computation for the feed-forward and backward process, "Extra" measures the extra computations used for guiding MTL, and "All" is the sum of them.

We ignore the complexity of sampling from a discrete distribution with $N$ elements, e.g. sampling an arm in MAB algorithm. What's more, we also ignore the optimization process in Nash-MTL and UW because they are quite efficient to compute. From the results in the Table 1, our method has moderate extra computation costs comparing with other methods, however, when considering the overall computation cost, our method achieves the lowest complexity because we only need to perform one feedforward-backward process which is the most time-consuming part during training. We provide detailed descriptions of these MTL methods in Appendix A.5.

## 4 Experiments

In this section, we conduct a comparative analysis between the proposed method against both single-task training (STL) and extensive multi-task learning (MTL) methods to demonstrate the efficacy of our approach in addressing various COPs under different evaluation criteria. Specifically, we examine two distinct scenarios: **(1)** Under identical training budgets, we aim to showcase the convenience of our method in automatically obtaining a universal combinatorial neural solver for multiple COPs, circumventing the challenges of balancing loss in MTL and allocating time for each task in STL; **(2)** Given the approximately same number of training epochs, we seek to illustrate that our method can derive a potent neural solver with excellent generalization capability. Furthermore, we employ the influence matrix to analyze the relationship between different COP types and the same COP type with varying problem scales.

We explore four types of COPs: the Travelling Salesman Problem (TSP), the Capacitated Vehicle Routing Problem (CVRP), the Orienteering Problem (OP), and the Knapsack Problem (KP). Detailed descriptions can be found in Appendix A.1. Three problem scales are considered for each COP: 20, 50, and 100 for TSP, CVRP, and OP; and 50, 100, and 200 for KP. We employ the notation "COP-scale", such as TSP-20, to denote a particular task, resulting in a total of 12 tasks. We emphasize that the derivation presented in Section 3.1 applies to a wide range of loss functions encompassing both supervised learning-based and reinforcement learning-based methods. In this study, we opt for reinforcement learning-based neural solvers, primarily because they do not necessitate manual labeling of high-quality solutions. As a representative method in this domain, we utilize the Attention Model (AM) (Kool et al., 2019) as the backbone and employ POMO (Kwon et al., 2020) to optimize its parameters. Concerning the bandit algorithm, we select Exp3 and the update frequency is set to 12 training batches. We discuss the selection of the MAB algorithms and update frequency in Appendix A.4, with details on the solving logic of neural solvers and training configurations in Appendix A.7.

### 4.1 Comparison under Same Training Budgets

We consider a practical scenario with limited training resources available for neural solvers for all tasks. Our method addresses this challenge by concurrently training all tasks using an appropriate task sampling strategy. However, establishing a schedule for STL is difficult due to the lack of information regarding

Table 2: Comparison among our proposed method, multi-task learning (MTL), and single task training (STL) utilizing the same training budget. Specifically, STLavg. and STLbal. denote the allocation of resources, with an even distribution and a balanced allocation ratio of $1:2:3$, respectively, among tasks with varying scales from small to large. The reported results depict the optimality gap ($\downarrow$) in the main aspects.

| | Method | TSP20 | TSP50 | TSP100 | CVRP20 | CVRP50 | CVRP100 | OP20 | OP50 | OP100 | KP50 | KP100 | KP200 | Avg. Gap |
|---|---|---|---|---|---|---|---|---|---|---|---|---|---|---|
| **Small Budget** | STL$_{\text{avg.}}$ | **0.009**% | 0.347% | 3.814% | 0.466% | 2.301% | 5.966% | −1.080% | 1.289% | 5.719% | **0.028**% | 0.014% | 0.017% | 1.574%±0.102 |
| | STL$_{\text{bal.}}$ | 0.021% | 0.341% | 3.176% | 0.608% | 2.312% | 4.775% | −0.970% | 1.288% | 4.799% | 0.034% | 0.014% | **0.016**% | 1.368%±0.138 |
| | Naive-MTL | 0.040% | 0.779% | 3.263% | 0.663% | 2.427% | 4.364% | −0.508% | 2.380% | 5.138% | 0.037% | 0.014% | 0.017% | 1.551%±0.086 |
| | UW | 0.039% | 0.496% | 2.353% | 0.464% | 1.929% | 3.742% | −0.434% | 2.268% | 4.819% | 0.035% | 0.015% | 0.017% | 1.312%±0.262 |
| | Bandit-MTL | 0.057% | 1.203% | 4.846% | 0.947% | 3.119% | 5.460% | −0.534% | 2.058% | 4.360% | 0.043% | 0.016% | 0.020% | 1.800%±0.584 |
| | PCGrad | 0.460% | 2.920% | 7.776% | 1.576% | 4.890% | 8.112% | 0.600% | 5.264% | 9.129% | 16.167% | 0.019% | 0.026% | 4.745%±2.360 |
| | CAGrad | 0.810% | 4.584% | 11.057% | 1.642% | 5.047% | 8.208% | 0.544% | 5.074% | 9.075% | 0.047% | 0.020% | 0.036% | 3.845%±0.692 |
| | Nash-MTL | 0.372% | 2.598% | 7.047% | 1.333% | 4.232% | 7.181% | 0.365% | 4.681% | 8.526% | 0.048% | 0.018% | 0.022% | 3.035%±0.964 |
| | TAG | 0.035% | 0.678% | 2.947% | 0.622% | 2.214% | 3.916% | −0.632% | 1.530% | 3.856% | 0.063% | 0.020% | 0.042% | 1.274%±0.096 |
| | Random | 0.060% | 0.577% | 3.029% | 0.608% | 2.233% | 4.117% | −0.891% | 1.061% | 3.483% | 0.041% | 0.014% | 0.021% | 1.196%±0.060 |
| | Ours | 0.024% | **0.262**% | **1.407**% | **0.385**% | **1.521**% | **2.798**% | **−1.081**% | **0.447**% | **1.883**% | 0.035% | **0.015**% | 0.018% | **0.643%±0.116** |
| **Median Budget** | STL$_{\text{avg.}}$ | **0.005**% | **0.182**% | 2.240% | 0.374% | 1.664% | 4.122% | **−1.156**% | 0.661% | 3.627% | 0.025% | 0.013% | 0.015% | 0.981%±0.074 |
| | STL$_{\text{bal.}}$ | 0.008% | 0.184% | 1.654% | 0.450% | 1.650% | 3.476% | −1.112% | 0.677% | 2.555% | 0.028% | 0.013% | 0.013% | 0.800%±0.022 |
| | Naive-MTL | 0.022% | 0.420% | 2.239% | 0.510% | 1.974% | 3.560% | −0.870% | 1.161% | 3.179% | 0.033% | 0.013% | 0.014% | 1.021%±0.034 |
| | UW | 0.031% | 0.304% | 1.796% | 0.392% | 1.621% | 3.180% | −0.741% | 1.370% | 3.770% | 0.030% | 0.013% | 0.016% | 0.982%±0.198 |
| | Bandit-MTL | 0.030% | 0.723% | 3.319% | 0.694% | 2.430% | 4.374% | −0.859% | 1.174% | 3.000% | 0.041% | 0.015% | 0.018% | 1.247%±0.404 |
| | PCGrad | 0.193% | 1.583% | 5.171% | 1.144% | 3.641% | 6.215% | 0.135% | 3.854% | 7.164% | 16.161% | 0.017% | 0.020% | 3.775%±4.050 |
| | CAGrad | 0.986% | 4.479% | 10.354% | 1.280% | 3.805% | 6.576% | 0.023% | 4.071% | 7.557% | 0.050% | 0.019% | 0.034% | 3.269%±1.152 |
| | Nash-MTL | 0.177% | 1.400% | 4.564% | 0.906% | 3.132% | 5.589% | −0.115% | 3.525% | 6.809% | 0.046% | 0.016% | 0.020% | 2.173%±0.650 |
| | TAG | 0.023% | 0.334% | 1.767% | 0.372% | 1.480% | 2.762% | −0.812% | 0.884% | 2.804% | 0.034% | 0.016% | 0.022% | 0.807%±0.074 |
| | Random | 0.044% | 0.290% | 1.779% | 0.452% | 1.706% | 3.310% | −1.050% | 0.818% | 2.028% | 0.030% | 0.011% | 0.014% | 0.786%±0.072 |
| | Ours | 0.015% | 0.192% | **0.955**% | **0.335**% | 1.226% | **2.260**% | −1.121% | **0.165**% | **1.127**% | **0.028**% | **0.011**% | **0.013**% | **0.434%±0.066** |
| **Large Budget** | STL$_{\text{avg.}}$ | **0.002**% | 0.117% | 1.290% | 0.284% | 1.278% | 3.086% | **−1.222**% | 0.249% | 2.101% | **0.018**% | 0.011% | 0.014% | 0.602%±0.012 |
| | STL$_{\text{bal.}}$ | 0.004% | **0.115**% | 1.020% | 0.367% | 1.284% | 2.588% | −1.176% | 0.255% | 1.592% | 0.024% | 0.011% | 0.012% | 0.508%±0.022 |
| | Naive-MTL | 0.015% | 0.229% | 1.300% | 0.395% | 1.611% | 2.901% | −0.965% | 0.447% | 1.910% | 0.027% | 0.013% | 0.012% | 0.658%±0.040 |
| | UW | 0.028% | 0.203% | 1.254% | 0.341% | 1.364% | 2.780% | −0.843% | 0.614% | 1.904% | 0.029% | 0.013% | 0.014% | 0.642%±0.020 |
| | Bandit-MTL | 0.008% | 0.231% | 1.113% | 0.489% | 1.366% | 2.513% | −1.201% | 0.419% | 1.975% | 0.035% | 0.013% | 0.018% | 0.581%±0.112 |
| | PCGrad | −0.006% | 0.412% | 3.688% | 0.625% | 2.250% | 4.033% | −0.878% | 1.938% | 3.960% | 15.176% | 0.015% | 0.019% | 2.603%±1.650 |
| | CAGrad | 0.194% | 1.971% | 6.947% | 1.363% | 2.802% | 4.622% | −0.611% | 2.494% | 5.848% | 0.055% | 0.025% | 0.024% | 2.145%±0.254 |
| | Nash-MTL | 0.049% | 0.658% | 1.943% | 0.502% | 2.005% | 3.478% | −0.985% | 1.642% | 3.742% | 0.045% | 0.018% | 0.021% | 1.093%±0.070 |
| | TAG | 0.006% | 0.058% | 1.193% | 0.312% | 1.287% | 2.253% | −0.929% | 0.594% | 2.092% | 0.029% | 0.013% | 0.015% | 0.577%±0.024 |
| | Random | 0.019% | 0.185% | 1.177% | 0.371% | 1.449% | 2.814% | −0.922% | 0.400% | 2.060% | 0.031% | 0.013% | 0.012% | 0.634%±0.028 |
| | Ours | 0.014% | 0.173% | **0.866**% | **0.321**% | **1.182**% | **2.165**% | −1.133% | **0.093**% | **0.987**% | 0.025% | **0.011**% | **0.011**% | **0.393%±0.056** |

resource allocation for each task, and MTL methods are hindered by efficiency issues arising from joint task training. In this section, we compare our method with naive STL and MTL methods in terms of the optimality gap: $gap\% = (1 - \frac{\text{obj.}}{\text{gt.}}) \times 100$, averaged over 10,000 instances for each task under an identical training time budget.

**Baselines** The total training time budget is designated as $B$. For STL framework, we consider three types of schedules for the allocation of time across varying problem scales: **(1)** Average allocation, denoted as STL$_{\text{avg.}}$, indicating a uniform distribution of resources for each task; **(2)** Balanced allocation with each type of COP receiving resources equitably for $\frac{B}{4}$, denoted as STL$_{\text{bal.}}$, signifying a size-dependent resource assignment with a 1:2:3 ratio from small to large problem scales, categorizing tasks into easy-median-hard levels; **(3)** Allocation determined by the bandit algorithm, denoted as STL$_{\text{bandit}}$, utilizes the selection ratio of each task after training the neural solver by the proposed method. The first schedule is suitable for realistic scenarios where information regarding the tasks is unavailable and the second is advantageous when prior knowledge is introduced. Although the third strategy is not typically available in practical applications, it serves to demonstrate the potential benefits of an adaptive training mechanism inherent in our proposed method. What's more, extensive MTL baselines are considered here: Bandit-MTL (Mao et al., 2021), PC-

Table 3: Training time per epoch, represented in minutes. The COPs are classified into three scales: small, median, and large, which correspond to the sizes of 20, 50, and 100, respectively (50, 100, and 200 for KP).

| COP | Small | Median | Large |
|---|---|---|---|
| TSP | 0.19 | 0.39 | 0.75 |
| CVRP | 0.27 | 0.50 | 0.90 |
| OP | 0.20 | 0.41 | 0.60 |
| KP | 0.34 | 0.61 | 1.10 |

Grad (Yu et al., 2020), Nash-MTL (Navon et al., 2022), Uncertainty-Weighting (UW) (Kendall et al., 2018), CAGrad (Liu et al., 2021a) and TAG (Fifty et al., 2021) (Detailed configurations for TAG are in Appendix A.6). We also involve the random policy which samples the task uniformly at each training slot, and the results of condensed version are presented in Table 2.

**Experimental settings** To mitigate the impact of extraneous computations, we calculate the time necessary to complete one epoch for each task and convert the training duration into the number of training epochs for STL. Utilizing the same device, the training time for for each task with STL and MTL methods can be found in Table 3 and Table 1. We assess three distinct training budgets: **(1)** Small budget: the time required to complete 500 training epochs using our method, approximately 1.59 days in GPU hours; **(2)** Medium budget: 1000 training epochs, consuming 3.28 days in GPU hours; and **(3)** Large budget: 2000 training epochs, spanning 6.64 days in GPU hours.

All methods were trained five times independently to ensure robust evaluation and statistical significance. Comprehensive experimental results are presented in Table 2. Due to horizontal space constraints, we report only the average performance values for individual tasks, while for the Average Gap across all tasks, we provide both the mean and two standard deviations across the five training runs. We refer the more detailed analysis of the stability of each method on individual tasks to Figure 11 in Appendix A.12.

**Results** In general, our method outperforms the MTL and STL methods in terms of averge gap across all the budgets used. When comparing with all MTL methods, our method demonstrates two superior advantages: **(1)** Better performance on the solution quality and efficiency: In Table 2, typical MTL methods fail to obtain a powerful neural solver efficiently, and some of them even work worse than naive MTL and STL in limited budgets. Further, as illustrated in Figure 2, it can be seen that the method proposed exhibits superior efficiency regarding its convergence speed and overall effectiveness; **(2)** More resources-friendly: The computation complexity of typical MTL methods grows linearly w.r.t. the number of tasks (a detailed analysis of the computational complexity of each MTL method is available in Appendix 3.4), conducting these training methods still needs heavy training resources (High-performance GPU with quite large memories). The exact training time for one epoch w.r.t. GPU hour are

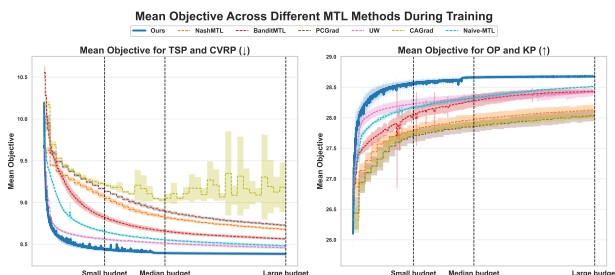

Figure 2: Comparative analysis of MTL methods during training: The left graph shows the mean objective function for TSP and CVRP (with a lower-is-better criterion), and the right graph shows the same for OP and KP (with a higher-is-better criterion), demonstrating the superior performance of our proposed method under varying computational budgets.

listed in Table 1. Under the same training setting, intermediate termination of prolonged training epoch for typical MTL methods incurs wasted computation resources. However, our method trains only one task at each time slot, resulting in rapid epoch-wise training that facilitates flexible experimentation and iteration.

As the training budgets increase, STL's advantages become evident in easier tasks such as TSP, CVRP-20, OP-20, and KP-50. However, our method continues to deliver robust results for more difficult tasks like CVRP-100 and OP-100. In addition to performance gains, the most notable advantage of our approach is that it does not require prior knowledge of the tasks and is capable of dynamically allocating resources for each task, which is crucial in real-world scenarios. When implementing STL, biases are inevitably introduced with equal allocation. As demonstrated in Table 2, the performance of two distinct allocation schedules can differ significantly: $\text{STL}_{\text{bal.}}$ consistently outperforms $\text{STL}_{\text{avg.}}$ due to the introduction of appropriate priors for STL. What's more, the random policy achieves competitive results compared to MTL baselines due to its efficient training strategy (sampling one task at a time). It slightly outperforms STL as our method(in small and median budgets) because positive transfer exists between same-type COP during the early stages of model training. However, unlike our method which selects tasks that maximize positive transfers across all tasks, the random policy doesn't actively explore and exploit task relationships. This lack of strategic task selection explains why it underperforms compared to our approach.

Table 4: The comparison results are obtained by training our model for 1000 epochs and STL models for 100 epochs each, amounting to a total of 1200 epochs.

| | TSP20 | TSP50 | TSP100 | CVRP20 | CVRP50 | CVRP100 | OP20 | OP50 | OP100 | KP50 | KP100 | KP200 | Avg. Gap |
|---|---|---|---|---|---|---|---|---|---|---|---|---|---|
| STL | **0.013**% | 0.234% | 1.673% | 0.490% | 1.613% | 3.342% | −1.156% | 0.812% | 2.678% | 0.030% | 0.014% | **0.012**% | 0.813% ± 0.070 |
| Ours | 0.015% | **0.192**% | **0.955**% | **0.335**% | **1.226**% | **2.260**% | **−1.121**% | **0.165**% | **1.127**% | **0.028**% | **0.011**% | 0.013% | **0.434**% ± 0.066 |

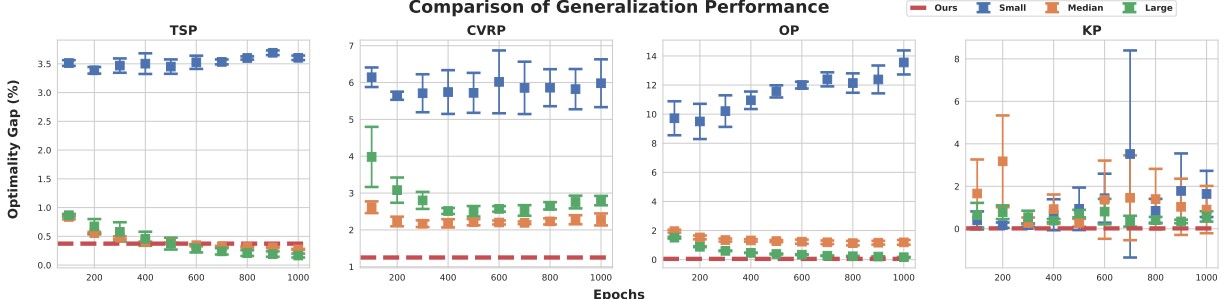

Figure 3: This figure compares the generalization performance across different scales for TSP, CVRP, OP, and KP. The y-axis represents the average test optimality gap (%) on small, median and large scales for models trained on small (blue), median (orange), and large (green) scale single-task datasets. The red line denotes the results of our method after training for 1000 epochs.

## 4.2 Comparative Analysis and Generalization Performance

In this part, we evaluated the performance of our method against single-task training (STL) under approximately equivalent total training epochs and generalization performance. Specifically, our method was trained for 1000 epochs across 12 tasks, while STL models were trained for 100 epochs per task, amounting to a total of 1200 epochs. The results are summarized in Table 4, highlighting the optimality gap for various tasks, including TSP, CVRP, OP, and KP. Compared to individual tasks, our method (trained 1000 epochs) consistently outperforms STL (trained $100 \times 12 = 1200$ epochs) across most tasks, with exceptions noted in TSP20, OP20, and KP50, indicating efficient utilization of the training epochs.

To assess the generalization performance, we compared our method, trained for 1000 epochs, against STL models trained individually for 1000 epochs. Figure 3 illustrates the average optimality gaps for TSP, CVRP, OP, and KP across different scales (small, median, and large). Our method demonstrates unparalleled superiority in three ways: **(1)** when considering the average performance on all problem scales for each type of COP, our method obtains the best results in CVRP, OP, and KP, and is equivalent to the results achieved by training TSP for about 500 epochs. This showcases our method's excellent generalization ability for problem scales; **(2)** Our method can handle various types of COPs under the same number of training epochs, which is impossible for STL due to the existence of task-specific modules; **(3)** Our method's training time is strictly shorter than the longest time-consuming task.

## 4.3 Study of the Influence Matrix

Our approach has an additional advantage as it facilitates the identification of the task relationship through the influence matrix developed in Section 3.2. The influence matrix enables us to capture the inherent relationships among tasks, providing empirical evidence based on experiences and observations within the learning-to-optimize community. A detailed view of the influence matrix is presented in Figure **??**, revealing significant observations: **(1)** Figure **??** highlights that the influence matrix computed using equation 10 possesses a diagonal-like block structure. This phenomenon suggests a strong correlation between the same type of COP, which is not present within different types of COPs due to the corresponding elements being insignificant. Furthermore, within the same type of COP, we observe that the effect of training a task on other tasks lessens with the increase in the difference of problem scales. Hence, training combinatorial neural solvers on one problem scale leads to higher benefits on similar problem scales than on those that are further

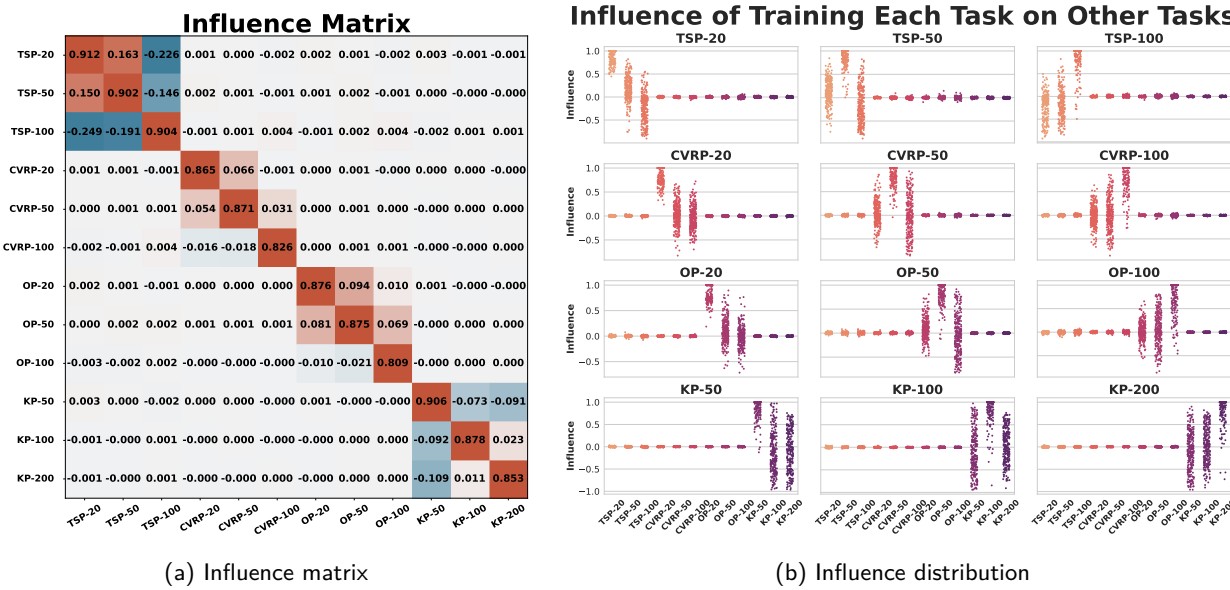

(a) Influence matrix

(b) Influence distribution

Figure 4: This figure provides a visual representation of the mutual influence between tasks. The left-hand side displays the average influence matrix, as defined in equation **??**, while the right-hand side illustrates the influence value throughout the training process.

away. Such scale generalization challenge for COPs has been noted in previous works, such as Figure 5 in Kool et al. (2019) and Figure 3 in Joshi et al. (2022), which demonstrate that neural solvers trained on specific problem sizes show degraded performance on other scales, with performance declining as the gap between training and testing sizes increases. This fact is in turn quantified by the influence matrix in our work. **(2)** Figure **??** presents a visualization of the influence resulting from equation 6, equation 7 over the course of the training process. Each point in the chart represents the influence of a particular task on another task at a specific time step. Notably, tasks belonging to the same type of COP are highly influential towards each other due to the large variance of their influence values. This phenomenon indicates that for different scales of the same COP type, both positive and negative transfer exist simultaneously. Positive transfer enables a trained neural solver to maintain basic solving capabilities compared to random initialization. However, the negative transfer arising from scale differences hinders the neural solver's generalization ability across different problem sizes. In contrast, influences between different types of COPs are negligible, evident from the influence values being concentrated around 0. This striking observation showcases that the employed combinatorial neural solver and algorithm , AM (Kool et al., 2019) and POMO (Kwon et al., 2020), segregate the gradient space into distinct orthogonal subspaces, and each of these subspaces corresponds to a particular type of COP. Furthermore, this implies that the gradient of training each variant of COP is situated on a low-dimensional manifold, motivating us to develop more parameter-efficient neural solver backbones and algorithms.

## 4.4 Results on Real Datasets

In this section, we evaluate the efficacy of the proposed method using real-world datasets: TSPLib (Reinelt, 1991) and CVRPLib (Uchoa et al., 2017). The training encompasses six distinct tasks: TSP20, TSP50, TSP100, CVRP20, CVRP50, and CVRP100. The experimental configurations adhere to those outlined in section 4.1, considering two budget scenarios: Small and Medium. From Table 2, we select the most effective baselines from Single-Task Learning (STL) and Multi-Task Learning (MTL), specifically $STL_{bal.}$ and UW (Kendall et al., 2018), respectively. The performance of $STL_{bal.}$ for a given instance is derived from the best outcomes across three models; for instance, the result for berlin52 in TSPLib is obtained from the optimal model among TSP20, TSP50, and TSP100. We only report the average results across the instances in Table 5 and detailed instance-wise results can be found in Appendix A.10.

Our experimental results demonstrate superior performance across both TSPLib and CVRPLib datasets. Under small budget constraints, our method achieves optimality gaps of 4.550% and 3.940% for TSPLib and CVRPLib respectively, consistently outperforming the baseline methods. The improvement is particularly significant compared to UW, which shows high variability (13.145% optimality gap for TSPLib small budget). Under median budget conditions, our approach maintains its advantage with the lowest optimality gaps (3.177% for TSPLib,

Table 5: Comparison of different methods across TSPLib and CVRPLib w.r.t optimality gap.

| | Small Budget | | Median Budget | |
|---|---|---|---|---|
| | TSPLib | CVRPLib | TSPLib | CVRPLib |
| STL$_{bal.}$ | 5.953% | 6.300% | 3.975% | 4.630% |
| UW | 13.145% | 5.895% | 6.277% | 5.105% |
| Ours | **4.550%** | **3.940%** | **3.177%** | **3.344%** |

3.344% for CVRPLib), demonstrating robust performance across different problem settings and budget constraints.

## 5 Conclusions and Futhre Works

In the era of large models, training a unified neural solver for multiple combinatorial tasks is in increasing demand, whereas such a training process can be prohibitively expensive. In this work, we propose an efficient training framework to boost the training of multi-task combinatorial neural solvers with a multi-armed bandit sampler. With this framework, we can efficiently obtain a unified neural solver capable of covering multiple types of COPs simultaneously. We believe that this framework can be powerful for multi-task learning in a broader sense, especially in scenarios where resources are limited, and generalization is crucial. It can also help analyze task relations in the absence of priors.

Furthermore, the proposed framework is model-agnostic, which makes it applicable to any existing neural solvers. We speculate that different neural solvers may produce varying results on the influence matrix, and a perfect neural solver may gain mutual improvements even from different types of COPs. Therefore, there is an urgent need to study the unified backbone and representation method for solving COPs.

## Acknowledgments

This work was supported by National Science and Technology Major Project under Grant 2022ZD0116408 and in part by the Shenzhen Agency of Innovation under Grant KJZD20240903095712016.

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

# A  Appendix

## A.1  Problem Description

**Traveling Salesman Problem (TSP)** The objective is to determine the shortest possible route that visits each location once and returns to the original location. In this study, we limit our consideration to the two-dimensional euclidean case, where the information for each location is presented as $(x_i, y_i) \in \mathbb{R}^2$ sampled from the unit square.

**Vehicle Routing Problem (VRP)** The Capacitated VRP (CVRP) (Toth & Vigo, 2014) consists of a depot node and several demand nodes. The vehicle begins and ends at the depot node, travels through multiple routes to satisfy all the demand nodes, and the total demand for each route must not exceed the vehicle capacity. The goal of the CVRP is to minimize the total cost of the routes while adhering to all constraints.

**Orienteering Problem (OP)** The Orienteering Problem (OP) is a variant of the Traveling Salesman Problem (TSP). Instead of visiting all the nodes, the objective is to maximize the total prize of visited nodes within a total distance constraint. Unlike the TSP and the Vehicle Routing Problem (VRP), the OP does not require selecting all nodes.

**Knapsack Problem (KP)** The Knapsack Problem strives to decide which items with various weights and values to be placed into a knapsack with limited capacity fully. The objective is to attain the maximum total value of the selected items while not surpassing the knapsack's limit.

## A.2  Loss Decomposition For Adam Optimizer

Adam optimizer (Kingma & Ba, 2015) is more widely used and popular in practice than standard gradient descent. Accordingly, we derive the loss decomposition for Adam optimizer in a manner consistent with the previous method. We first summarize the update rule of Adam as follows:

$$\Theta(t) = \Theta(t-1) - \alpha \frac{\sqrt{\sum_{\tau=1}^{t-1} \beta_2^{t-\tau}}}{\sum_{\tau=1}^{t-1} \beta_1^{t-\tau}} \frac{\sum_{\tau=1}^{t} \beta_1^{t-\tau} g(\tau)}{\sqrt{\sum_{\tau=1}^{t} \beta_2^{t-\tau} ||g(\tau)||^2 + \epsilon}} = \Theta(t-1) - \eta_t \sum_{\tau=1}^{t} \beta_1^{t-\tau} g(\tau)$$

where $g(\tau) = \nabla L(\Theta(\tau-1))$ and $g_0 = \mathbf{0}, \eta_t = \frac{\sqrt{\sum_{\tau=1}^{t-1} \beta_2^{t-\tau}}}{\sum_{\tau=1}^{t-1} \beta_1^{t-\tau}} \frac{\alpha}{\sqrt{\sum_{\tau=1}^{t} \beta_2^{t-\tau} ||g_\tau||^2 + \epsilon}}, \beta_i, i = 1, 2$ are exponential average parameters for the first and second order gradients. Our assumption is that sharing the second moment term correction for all tasks can be easily implemented by using a single optimizer during training.

Given that the update is predicated on the optimization trajectory's history, we can use comparable calculations in gradient descent to infer Adam's contribution breakdown. Starting at the same point:

$$\mathbb{1}(a_t = T_q^p)\Delta L_j^i(t, t+1) = \begin{cases} -\eta_t \nabla^T L_j^i(\Psi^i(t)) \sum_{\tau=1}^{t+1} \beta_1^{t+1-\tau} g(\tau) & \text{if } p = i \\ -\eta_t \nabla_{\theta^{\text{share}}}^T L_j^i(\Psi^i(t)) \sum_{\tau=1}^{t+1} \beta_1^{t+1-\tau} g_{\text{share}}(\tau) & \text{Otherwise} \end{cases} \tag{11}$$

With a little abuse of notation, $g(\tau)$ means the gradient w.r.t. the selection task at time slot $\tau$ and $g_{\text{share}}(\tau)$ is the version of taking derivatives only one shared parameters. Then plugging equation 11 into equation 3,

we have

$$
\begin{aligned}
L_j^i(\Theta^i(t_2)) - L_j^i(\Theta^i(t_1)) = &-\underbrace{(\nabla^T L_j^i(\Psi^i(t_1)) \sum_{t=t_1}^{t_2} \mathbb{1}(a_t = T_j^i) \eta_t \sum_{k=1}^{t} \beta_1^{t-k} g(k-1))}_{(a)\ \text{effects of training task } T_j^i:\ e_j^i(t_1,t_2)} \\
&+ \underbrace{\nabla^T L_j^i(\Psi^i(t_1)) \sum_{\substack{q=1 \\ q \neq j}}^{n_i} \sum_{t=t_1}^{t_2} \mathbb{1}(a_t = T_q^i) \eta_t \sum_{k=1}^{t} \beta_1^{t-k} g(k-1)}_{(b)\ \text{effects of training task } \{T_q^i, q \neq j\}:\ \{e_q^i((t_1,t_2)), q \neq j\}} \\
&+ \underbrace{\nabla_{\theta^{\text{share}}}^T L_j^i(\Psi^i(t_1)) \sum_{\substack{p=1 \\ p \neq i}}^{K} \sum_{q=1}^{n_p} \sum_{t=t_1}^{t_2} \mathbb{1}(a_t = T_q^p) \eta_t \sum_{k=1}^{t} \beta_1^{t-k} g_{\text{share}}(k-1))}_{(c)\ \text{effects of training task } \{T_q^p, p \neq i\}:\{e_q^p(t_1,t_2), q=1,2,\ldots,n_p, p \neq i\}},
\end{aligned}
\tag{12}
$$

Three similar parts are obtained finally.

### A.3 Proof and Discussion on Theorem 1

**Proof 2 (Proof of theorem 1:)**

$$
\begin{aligned}
&|\hat{m}_j^i(T_q^p; t_1, t_2) - \tilde{m}_j^i(T_q^p; t_1, t_2)| \\
&= \left| \sum_{t=t_!}^{t_2} \frac{\mathbb{1}(a_t = T_q^p)}{\sum_{t'=t_!}^{t_2} \mathbb{1}(a_{t'} = T_q^p)} \cdot \left( \frac{\nabla L_j^i(\Theta^i(t))}{||\nabla L_j^i(\Theta^i(t))||} - \frac{\nabla L_j^i(\Theta^i(\tau_{i,j}(t)))}{||\nabla L_j^i(\Theta^i(\tau_{i,j}(t)))||} \right)^T \frac{\nabla L_q^p(t)}{||\nabla L_q^p(t)||} \right| \\
&\leq \sum_{t=t_!}^{t_2} \frac{\mathbb{1}(a_t = T_q^p)}{\sum_{t'=t_!}^{t_2} \mathbb{1}(a_{t'} = T_q^p)} \cdot \left| \left( \frac{\nabla L_j^i(\Theta^i(t))}{||\nabla L_j^i(\Theta^i(t))||} - \frac{\nabla L_j^i(\Theta^i(\tau_{i,j}(t)))}{||\nabla L_j^i(\Theta^i(\tau_{i,j}(t)))||} \right)^T \frac{\nabla L_q^p(t)}{||\nabla L_q^p(t)||} \right| \\
&\leq \sum_{t=t_!}^{t_2} \frac{\mathbb{1}(a_t = T_q^p)}{\sum_{t'=t_!}^{t_2} \mathbb{1}(a_{t'} = T_q^p)} \left\| \frac{\nabla L_j^i(\Theta^i(t))}{||\nabla L_j^i(\Theta^i(t))||} - \frac{\nabla L_j^i(\Theta^i(\tau_{i,j}(t)))}{||\nabla L_j^i(\Theta^i(\tau_{i,j}(t)))||} \right\| \\
&\overset{(*)}{=} \sum_{t=t_!}^{t_2} \frac{\mathbb{1}(a_t = T_q^p)}{\sum_{t'=t_!}^{t_2} \mathbb{1}(a_{t'} = T_q^p)} \frac{\left\| \nabla L_j^i(\Theta^i(t)) - \nabla L_j^i(\Theta^i(\tau_{i,j}(t))) \right\|}{G} \\
&\overset{(**)}{=} \sum_{t=t_!}^{t_2} \frac{\mathbb{1}(a_t = T_q^p)}{\sum_{t'=t_!}^{t_2} \mathbb{1}(a_{t'} = T_q^p)} \frac{\left\| \nabla^2 L_j^i(\Psi^i) \sum_{t'=\tau_{i,j}(t)+1}^{t} \eta_{t'} \nabla L_{n(t')}^{m(t')}(\Theta^{m(t')}) \right\|}{G} \\
&\leq \sum_{t=t_!}^{t_2} \frac{\mathbb{1}(a_t = T_q^p)}{\sum_{t'=t_!}^{t_2} \mathbb{1}(a_{t'} = T_q^p)} \frac{||\nabla^2 L_j^i(\Psi^i)|| \sum_{t'=\tau_{i,j}(t)+1}^{t} \eta_{t'} ||\nabla L_{n(t')}^{m(t')}(\Theta^{m(t')})||}{G} \\
&\overset{(***)}{\leq} \frac{M}{\sum_{t'=t_!}^{t_2} \mathbb{1}(a_{t'} = T_q^p)} \sum_{t=t_!}^{t_2} \mathbb{1}(a_t = T_q^p) \sum_{t'=\tau_{i,j}(t)+1}^{t} \eta_{t'},
\end{aligned}
$$

*where $(*)$ holds because we suppose all gradients have the same norm $G$ during the training which can be guaranteed by regularizing the gradients; $(**)$ holds due to the Taylor expansion for $\nabla L_j^i(\Theta^i(t))$ on $\Theta^i(\tau_{i,j}(t))$ and $L_{n(t')}^{m(t')}$ denotes the loss for training task $T_{n(t')}^{m(t')}$ selected at time step $t'$; $(***)$ is obtained from the assumption on the bounded Hession of the loss function. Then let $\eta^{max}(t_1, t_2) = \max_{t=t_1,\ldots,t_2}\{\eta_t\}, c_q^p(t_1, t_2) = \sum_{t=t_!}^{t_2} \mathbb{1}(a_t = T_q^p)$, the final result is obtained:*

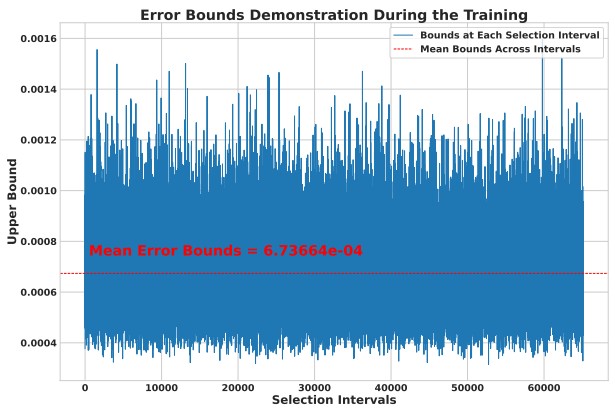
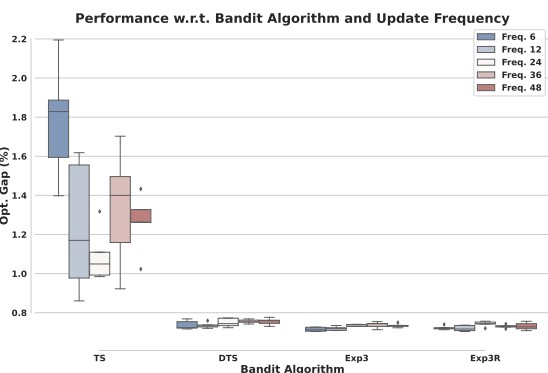

Figure 5: This figure showcases the fluctuation of mean error bounds during the training process. The red dashed line is the mean error bounds computed across all intervals.

Figure 6: This figure presents the comparison results of various bandit algorithms and update frequencies in terms of optimality gap (%).

$$|\hat{m}_j^i(T_q^p; t_1, t_2) - \tilde{m}_j^i(T_q^p; t_1, t_2)| \leq \frac{M\eta^{max}(t_1, t_2)}{c_q^p(t_1, t_2)} \sum_{t=t_1}^{t_2} \mathbb{1}(a_t = T_q^p)(t - \tau_{i,j}(t)).$$

In practice, we can easily verify the magnitude of $\frac{\eta^{\max}(t_1,t_2)}{c_q^p(t_1,t_2)} \sum_{t=t_1}^{t_2} \mathbb{1}(a_t = T_q^p)(t - \tau_{i,j}(t))$ during the training process. Following the experimental setup in the main paper, with $\eta^{\max} = \eta = 10^{-4}$, and a selection frequency of 12, we calculate the average of this error upper bound for all tasks within a selection cycle, i.e., $\frac{\eta^{\max}(t_1,t_2)}{\sum_{i,j,p,q} 1} \sum_{i,j,p,q} \sum_{t=t_1}^{t_2} \frac{\mathbb{1}(a_t = T_q^p)(t - \tau_{i,j}(t))}{c_q^p(t_1,t_2)}$. The experimental results are demonstrated in Figure 5, showing that the average error upper bound steadily maintains at the magnitude of $10^{-4}$. Regarding the upper bound of the norm of the Hessian matrix in neural networks, there have been numerous prior studies (Sagun et al., 2017; Ghorbani et al., 2019; Yao et al., 2020; Li et al., 2020) investigating this topic. Specifically, Sagun et al. (2017) pointed out that in the late stage of neural network optimization, most of the eigenvalues of its Hessian matrix are near zero; Ghorbani et al. (2019) found that the large negative eigenvalues of the Hessian disappeared rapidly, with the overall spectrum shape stabilizing in only very few training steps; Similarly, Li et al. (2020) discovered that as training progresses, the larger eigenvalues of the neural network's Hessian gradually decrease and concentrate near zero. These findings have been validated on various image datasets and neural network architectures. Although there are currently no specific observations for COP tasks, we speculate that similar conclusions for small value of $M$ might hold.

## A.4 Discussion on The Bandit Algorithm and Update Frequency

As shown in equation 2, the effect of the training task $T_q^p$ on $T_j^i$ can be computed as

$$\mathbb{1}(a_t = T_q^p) \nabla^T L_j^i(\Psi^i(t)) \nabla L_q^p(\Theta^p(t)).$$

This is subject to the indicator function $\mathbb{1}(a_t = T_q^p)$, which determines whether the task $T_q^p$ is selected at time step $t$. We first highlight the following tips: **(1)** For the stability and accuracy of the gradients, it is recommended to involve more than one step in the process of collecting gradient information. However, having an overly slow update frequency may yield incorrect results due to the lazy update of the bandit algorithm; **(2)** When the update frequency is larger than 1, UCB family algorithms are unsuitable as they tend to greedily select the same task in the absence of updates. Therefore, the update frequency is a crucial hyper-parameter to specify, and Thompson Sampling and adversary bandit algorithms are suitable in this framework due to their higher level of randomness.

Based on above discussions, we present empirical evidence and elaborate on the details. We performed experiments for the 12 tasks under small budgets, with five repetitions each. Five update frequencies were considered: 6, 12, 24, 36, and 48. The performances w.r.t. optimality gap are presented in Figure 6.

**Effects of bandit algorithms** The four algorithms considered are: Exp3, Thompson Sampling (TS), Exp3R, and Discounted Thompson Sampling (DTS). They have more exploration characteristics than UCB family algorithms with update delays. Moreover, Exp3R and DTS have the capability to handle changing environments. According to Figure 6, TS performs the worst among these four algorithms, as it fails to handle potential adversaries and changing environments. DTS performs more robustly than TS since it involves a discounted factor. Exp3 and Exp3R provide good results because they are able to handle adversaries and detect environmental changes. However, Exp3R does not perform significantly better than Exp3 due to the neural solver's gradual and slow improvement, resulting no abrupt changes for Exp3R to detect. Based on the observed performance, it appears that simple procedures such as introducing a discounted factor in DTS and basic adversary bandit algorithms such as Exp3 are sufficient for handling our case.

**Effects of update frequency** The update frequency affects the accuracy of influence information approximation and the tension in the bandit algorithm. Appropriate selections must balance these two factors. Figure 6 shows that the frequency of 12 generally yields the best results across different bandit algorithms. DTS and Exp3 exhibit deteriorating performance with higher frequencies, resulting from numerous lazy updates. By contrast, Exp3R does not have this property because increasing the frequency helps detect changing points more quickly. As a consequence, the number of tasks (12 in our case) appears to be an appropriate empirical choice to balance these two factors.

### A.5 Details of Multi-task Learning Methods

In this section, we provide detailed descriptions of the MTL baselines used in our main experiments. Specifically, we elaborate on the algorithm-level details of Bandit-MTL (Mao et al., 2021), PCGrad (Yu et al., 2020), Nash-MTL (Navon et al., 2022), Uncertainty-Weighting (UW) (Kendall et al., 2018), CAGrad (Liu et al., 2021a) and IMTL (Liu et al., 2021b). All these methods train multiple tasks simultaneously at each training step and achieve positive transfer by designing different strategies to balance task-specific losses through loss weights. Therefore, these methods share a basic computational complexity of $\mathcal{O}(N(F + B))$ for network forward and backward propagation in one training step, where $F$ and $B$ denote the computational cost of forward and backward pass respectively. We also present the detailed derivation of their computational complexity as shown in Table 1.

**Bandit-MTL** Bandit-MTL formulates the task weighting problem as an adversarial multi-armed bandit problem. At each training step, it computes the loss weights by regularizing the variance of task losses, which means it attempts to find weights that can both minimize the weighted sum of task losses and reduce the variance among different task losses.These weights are optimized using mirror gradient ascent under the constraint that they sum to one. Since the computation of loss weights only depends on the task losses and the optimization process is efficient in practice, Bandit-MTL introduces an additional $\mathcal{O}(N)$ computational complexity for updating the weights.

**PCGrad** PCGrad addresses the gradient conflict issue in multi-task learning through gradient surgery. At each training step, it identifies conflicting gradients between task pairs and modifies them by projecting each gradient onto the normal plane of the other, effectively removing the interfering components. Since PCGrad needs to compute pair-wise gradient projections for the gradients of model parameters, it introduces an additional computational complexity of $\mathcal{O}(N^2D)$ at most, where $D$ is the number of model parameters.

**Nash-MTL** Nash-MTL reformulates the gradient combination in multi-task learning as a cooperative bargaining game, where each task is treated as a player negotiating for an agreed update direction. It utilizes the Nash bargaining solution from game theory to find a proportionally fair update direction that benefits all tasks without being dominated by any single large gradient. To compute this solution, Nash-MTL requires calculating pairwise gradient inner products and solving a linear system using a variation of the concave-convex procedure. Although the optimization process is efficient in practice, the computation

of pairwise gradient operations introduces an additional computational complexity of $\mathcal{O}(N^2 D)$, where D is the number of model parameters.

**UW** UW proposes a principled approach to balance multiple task losses by leveraging homoscedastic uncertainty. It interprets the homoscedastic uncertainty as task-dependent weighting and uses this interpretation to automatically adjust the relative weights of different tasks. The method derives these weights directly from the estimated noise levels of each task, making it applicable to both regression and classification tasks. Since the weights can be directly computed from given noise levels and applied to the losses, UW introduces only $\mathcal{O}(1)$ additional computational complexity.

**CAGrad** CAGrad proposes a novel approach to balance multiple tasks by considering both the average loss and the worst-case local improvement of individual tasks. At each training step, it seeks an update direction that maximizes the minimum improvement across all tasks within a neighborhood of the average gradient, thereby ensuring that no single task is disproportionately disadvantaged. Due to the complexity of solving the underlying min-max optimization problem, the explicit computational complexity analysis of CAGrad is not provided in our study.

**IMTL** IMTL aims to achieve fair optimization across tasks through two key components: loss balancing and gradient balancing. For task-shared parameters, it computes scaling factors through a closed-form solution to ensure that the aggregated gradient has equal projections onto individual task gradients. For task-specific parameters, it dynamically adjusts task loss weights to maintain comparable loss scales across tasks. The method involves two stages of computation: the loss balance stage (IMTL-L) which requires normalized gradients computation for each task, and the gradient balance stage which involves matrix operations for computing scaling factors. Due to these computations, IMTL introduces an additional computational complexity of $\mathcal{O}(N^2 D + N^3)$, where the $N^2 D$ term comes from matrix multiplications between task gradients, and the $N^3$ term arises from solving a $(N-1) \times (N-1)$ linear system.

### A.6 Configurations and Detailed Results of TAG

Strictly speaking, TAG (Fifty et al., 2021) is not suitable as a baseline because it involves two stages: collecting task affinity and training. The main challenges lie in how to allocate weights to these two stages under limited resources and how to determine the optimal number of groups. However, to highlight the superiority of the proposed method, we disregard the time required for collecting task affinity and allocate training time proportionally to the number of tasks in each group. The number of groups is set to 2, 3, and 4, and the best-performing result is ultimately selected. The grouping results obtained from TAG are demonstrated in Table 6. As evident from the table, the results of TAG are comparable to those of MTL-based methods under all budget constraints and grouping strategies, both of which are significantly inferior to our proposed approach.

### A.7 Experimental Settings

**Solving Logic of POMO** POMO (Kwon et al., 2020) applies the construction scheme to solve combinatorial optimization problems (COPs) by generating a feasible solution for an instance incrementally. For example, in the Travelling Salesman Problem (TSP), when solving an instance with n nodes, if the partial solution currently contains k nodes, one node needs to be selected from the remaining n - k nodes. Once a specific node is selected and added to the partial solution, the next partial solution, containing k + 1 nodes, is determined. This process continues until all nodes are included in the partial solution, at which point a feasible solution is achieved.

**Model structure** We adopt the same model structures as in POMO (Kwon et al., 2020) to build our model. To train various COPs in a unified model, we use a separate MLP on top of the model for each problem, which we call *Header*. This header facilitates correlation of input features with different dimensions. For TSP, we use two-dimensional coordinates, $\{(x_i, y_i), i = 1, 2, ..., n\}$, as input, while CVRP and OP have additional constraints on customer demand and vehicle capacity, in addition to two-dimensional coordinates. Hence, their input dimensions are 3 and 3, respectively. Moreover, in OP, the prize is assigned based on the

Table 6: Performance of TAG under different grouping strategies. The reported results depict the optimality gap ($\downarrow$) in the main aspects.

| Budget | Group | TSP20 | TSP50 | TSP100 | CVRP20 | CVRP50 | CVRP100 | OP20 | OP50 | OP100 | KP50 | KP100 | KP200 | Avg. Gap |
|---|---|---|---|---|---|---|---|---|---|---|---|---|---|---|
| Small Budget | 1 | - | - | - | 0.624% | 2.196% | 3.965% | - | - | - | - | - | - | 1.270% |
| | 2 | 0.033% | 0.663% | 2.857% | 0.630% | 2.303% | 4.103% | −0.622% | 1.542% | 3.882% | 0.059% | 0.020% | 0.039% | |
| | 1 | 0.015% | 0.237% | - | - | - | - | - | - | - | - | - | - | 1.401% |
| | 2 | - | - | - | 0.648% | 2.317% | 4.182% | - | - | - | - | - | - | |
| | 3 | 0.038% | 0.737% | 3.031% | 0.660% | 2.402% | 4.209% | −0.553% | 1.720% | 4.155% | 0.045% | 0.015% | 0.018% | |
| | 1 | 0.021% | 0.270% | - | - | - | - | - | - | - | - | - | - | 1.370% |
| | 2 | 0.050% | 0.443% | 2.179% | - | - | - | - | - | - | - | - | - | |
| | 3 | - | - | - | 0.678% | 2.377% | 4.322% | - | - | - | - | - | - | |
| | 4 | 0.046% | 0.792% | 3.274% | 0.703% | 2.468% | 4.344% | −0.516% | 2.010% | 4.649% | 0.037% | 0.014% | 0.020% | |
| Median Budget | 1 | - | - | - | 0.379% | 1.496% | 2.787% | - | - | - | - | - | - | 0.754% |
| | 2 | 0.025% | 0.310% | 1.745% | 0.459% | 1.820% | 3.285% | −0.812% | 0.896% | 2.590% | 0.032% | 0.016% | 0.023% | |
| | 1 | 0.008% | 0.141% | - | - | - | - | - | - | - | - | - | - | 0.755% |
| | 2 | - | - | - | 0.400% | 1.577% | 2.915% | - | - | - | - | - | - | |
| | 3 | 0.031% | 0.368% | 1.975% | 0.482% | 1.883% | 3.429% | −0.794% | 0.957% | 2.736% | 0.034% | 0.015% | 0.018% | |
| | 1 | 0.012% | 0.186% | - | - | - | - | - | - | - | - | - | - | 0.849% |
| | 2 | 0.028% | 0.212% | 1.218% | - | - | - | - | - | - | - | - | - | |
| | 3 | - | - | - | 0.447% | 1.741% | 3.199% | - | - | - | - | - | - | |
| | 4 | 0.035% | 0.452% | 2.378% | 0.555% | 2.063% | 3.667% | −0.700% | 1.134% | 3.092% | 0.035% | 0.014% | 0.019% | |
| Large Budget | 1 | - | - | - | 0.301% | 1.174% | 2.212% | - | - | - | - | - | - | 0.566% |
| | 2 | 0.022% | 0.210% | 1.212% | 0.352% | 1.518% | 2.797% | −0.910% | 0.543% | 1.919% | 0.032% | 0.013% | 0.018% | |
| | 1 | 0.004% | 0.078% | - | - | - | - | - | - | - | - | - | - | 0.562% |
| | 2 | - | - | - | 0.310% | 1.207% | 2.260% | - | - | - | - | - | - | |
| | 3 | 0.020% | 0.211% | 1.251% | 0.368% | 1.521% | 2.785% | −0.913% | 0.556% | 1.979% | 0.031% | 0.013% | 0.014% | |
| | 1 | 0.006% | 0.093% | - | - | - | - | - | - | - | - | - | - | 0.581% |
| | 2 | 0.009% | 0.109% | 0.667% | - | - | - | - | - | - | - | - | - | |
| | 3 | - | - | - | 0.328% | 1.337% | 2.509% | - | - | - | - | - | - | |
| | 4 | 0.024% | 0.264% | 1.443% | 0.420% | 1.670% | 3.040% | −0.853% | 0.654% | 2.163% | 0.037% | 0.014% | 0.013% | |

distance between the node and the depot node, following the setting in AM (Kool et al., 2019). The KP takes two-dimensional inputs, $\{(w_i, v_i), i = 1, 2, ..., n\}$, with $w_i$ and $v_i$ representing the weight and value of each item, respectively. As such, we introduce four kinds of *Header* to embed features with different dimensions to 128. The embeddings obtained from the *Header* are then passed through a shared *Encoder*, composed of six encoder layers based on the Transformer (Vaswani et al., 2017). Finally, we employ four type-specific *Decoder*s, one for each COP, to make decisions in a sequential manner. The shared *Encoder* has the bulk of the model's capacity because the *Header* and *Decoder* are lightweight 1-layer MLPs. Furthermore, when solving a specific COP, we only need to use the relevant *Encoder*, *Header*, and *Decoder* for evaluation. Since the model size is precisely the same, the inference time required is similar to that of single-task learning.

**Hyperparameters** In each epoch, we process a total of 100×1000 instances with a batch size of 512. The POMO size is equal to the problem scale, except for KP-200, where it is 100. We optimize the model using Adam (Kingma & Ba, 2015) with a learning rate of 1e-4 and weight decay of 1e-6. The training of the model involves 1000 epochs in the standard setting. The learning rate is decreased by 1e-1 at the 900th epoch. During the first epoch, we use the bandit algorithm to explore at the beginning of the training process. We then collect gradient information by updating the bandit algorithm with every 12 batches of data. The model is trained using 8 Nvidia Tesla A100 GPUs in parallel, and the evaluations are done on a single NVIDIA GeForce RTX 3090.

**Bandit settings** We utilized the open-source repository (Besson, 2018) for implementing the bandit algorithms in this study with default settings.

## A.8 Loss and Gradient Norm of Each Task

One intuitive method of measuring the effect of training is to calculate the ratio of losses between adjacent training sessions. These ratios can be used to calculate training rewards for each corresponding task. How-

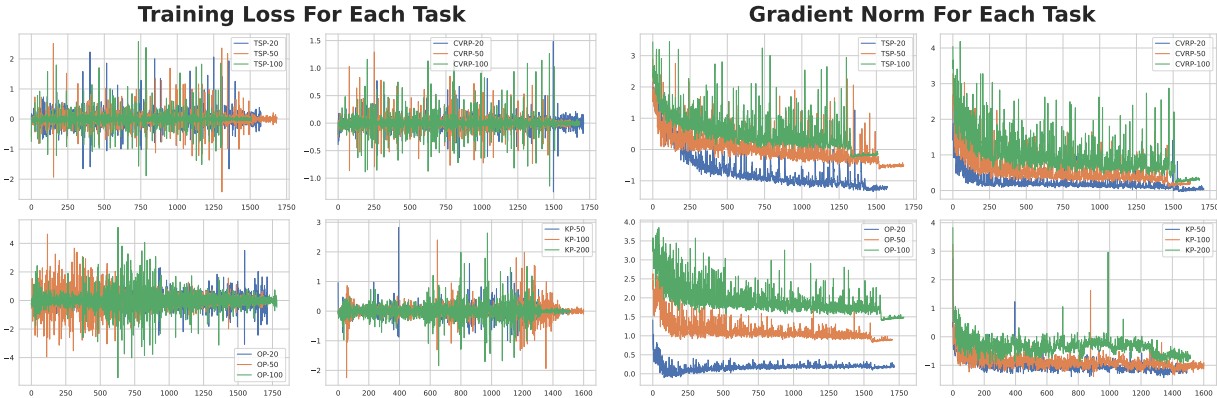

Figure 7: Training loss for each task.          Figure 8: Gradient norm for each task.

ever, as shown in Figure 7, this method of calculating rewards is not effective because they are not sufficiently distinct to guide the training process properly.

Computing the inner products of corresponding gradients to analyze how training one task affects the others can lead to a misleading calculation of rewards and training process. Figure 8 visualizes gradient norms for each task in the logarithmic scale. We observe that the gradient norms are not in the same scale, which becomes problematic when jointly training different COP types. In such cases, the rewards of certain COP types (such as CVRP in our experiments) may dominate the rewards of other types.

## A.9 Clarification on Performance Improvements and Further Ablations

We would like to clarify that cross-problem-type performance improvements do not stem from the shared encoder learning mutual improvement information. Rather, the improvements come from efficient training (see GPU hours/epoch in Table 3) and positive transfers in independent model parameter subspaces for each COP type: As observed through the influence matrix in Figure **??**, different COP types correspond to nearly orthogonal subspaces in the model parameter space. Each subspace learns solution information for its corresponding COP type, though this allocates fewer training parameters to each COP type. Based on this understanding, we further enhanced our method by the following setting.

Given a neural solver backbone and considering that different types of COPs exhibit minimal correlation, we investigate the effectiveness of our method by training separately on each COP type. Following the experimental setup in Table 2, we compared two training approaches: (1) class-specific training (Ours-4G), which groups 12 tasks into four problem classes (TSP, CVRP, OP, and KP) with the budget equally divided among these classes, and (2) collective training of all 12 tasks (Ours-12T). The comparison was conducted across three budget levels: Small, Medium, and Large.

Table 7: Comparison of performance between Ours-12T and Ours-4G under different budget sizes

| Budget | Method | TSP20 | TSP50 | TSP100 | CVRP20 | CVRP50 | CVRP100 | OP20 | OP50 | OP100 | KP50 | KP100 | KP200 | Avg. Gap |
|---|---|---|---|---|---|---|---|---|---|---|---|---|---|---|
| Small | Ours-12T | 0.019% | 0.248% | 1.325% | 0.373% | 1.476% | 2.741% | -1.107% | 0.402% | 1.826% | 0.033% | 0.014% | 0.020% | 0.614% |
| | Ours-4G | 0.013% | 0.167% | 0.919% | 0.342% | 1.323% | 2.444% | -1.025% | 0.602% | 2.126% | 0.045% | 0.017% | 0.017% | **0.583%** |
| Medium | Ours-12T | 0.014% | 0.195% | 0.911% | 0.331% | 1.199% | 2.219% | -1.138% | 0.119% | 1.040% | 0.029% | 0.012% | 0.013% | 0.412% |
| | Ours-4G | 0.008% | 0.134% | 0.697% | 0.295% | 1.131% | 2.105% | -1.126% | 0.193% | 1.373% | 0.047% | 0.029% | 0.019% | **0.409%** |
| Large | Ours-12T | 0.013% | 0.181% | 0.842% | 0.315% | 1.156% | 2.122% | -1.135% | 0.041% | 0.901% | 0.025% | 0.011% | 0.011% | 0.374% |
| | Ours-4G | 0.005% | 0.077% | 0.427% | 0.281% | 0.967% | 1.823% | -1.201% | 0.034% | 0.971% | 0.045% | 0.018% | 0.015% | **0.288%** |

Table 7 presents the performance comparison between class-specific training (Ours-4G) and collective training (Ours-12T) under different budget sizes. Across all three budget levels, Ours-4G consistently outperforms the best results of Ours-12T among 5 repeated runs, achieving lower average optimality gaps (0.583% vs. 0.614% for small budget, 0.409% vs. 0.412% for medium budget, and 0.288% vs. 0.374% for large budget).

This superior performance of Ours-4G demonstrates that when task relationships are known a priori (e.g., through the influence matrix), separating tasks into groups can lead to better results. This improvement may be attributed to reduced interference from less correlated tasks, allowing the model to focus more effectively on training highly correlated tasks together. However, it's worth noting that Ours-12T, which trains all tasks collectively without requiring any prior knowledge of task relationships, exhibits more general applicability while still achieving competitive results as shown in Table 2. These results highlight the complementary advantages of our proposed methods: Ours-4G achieves optimal performance when task relationships are known, while Ours-12T provides a robust and effective solution when no prior knowledge is available, making our framework adaptable to various practical scenarios.

## A.10    Detailed Results on TSPLib and CVRPLib

In this section, we present the experimental results on TSPLib and CVRPLib that were omitted in the main text.

The results presented in Table 8 clearly illustrate the superiority of our proposed method across both budget scenarios on the TSPLib dataset. Notably, our method consistently achieves the lowest optimality gaps, highlighting its robust performance under varying budget constraints. The average optimality gap for our method is 4.55% and 3.18% for small and median budgets respectively, which are significantly lower compared to the other approaches, including $STL_{bal.}$ and UW. Moreover, an analysis of instances with problem scales of 100 or greater reveals that our method not only maintains but often enhances its performance advantage as the problem scale increases. This trend underscores our method's excellent scalability and generalization across larger problem sizes, which is critical for practical applications where larger datasets are common.

Table 8: Comparison results on TSPLib. The reported results depict the optimality gap (%) (↓) in the main aspects.

| Instances | Small Budget | | | Median Budget | | |
|---|---|---|---|---|---|---|
| | $STL_{bal.}$ | UW | Ours | $STL_{bal.}$ | UW | Ours |
| berlin52 | **0.077**% | 11.849% | 0.988% | 0.065% | 3.161% | **0.0**% |
| st70 | **0.997**% | 3.125% | 1.024% | 0.837% | 1.258% | **0.35**% |
| pr76 | 0.828% | 2.575% | **0.818**% | **0.032**% | 1.552% | 0.69% |
| eil76 | 2.61% | 4.565% | **2.39**% | **1.237**% | 1.978% | 1.894% |
| rd100 | 1.877% | 9.044% | **0.568**% | **0.665**% | 0.855% | 0.673% |
| kroA100 | 4.887% | 11.177% | **3.229**% | 1.907% | 4.494% | **0.772**% |
| kroC100 | **1.269**% | 12.014% | 1.564% | 1.508% | 6.239% | **0.292**% |
| kroD100 | 4.223% | 12.277% | **2.352**% | 2.528% | 5.378% | **1.699**% |
| eil101 | 2.966% | 7.203% | **2.379**% | 2.379% | 3.526% | **1.343**% |
| lin105 | 3.727% | 15.204% | **3.449**% | 3.873% | 7.177% | **3.775**% |
| ch130 | 4.7% | 8.739% | **2.783**% | 2.689% | 2.899% | **1.448**% |
| ch150 | 4.715% | 12.064% | **3.295**% | 2.539% | 3.739% | **2.432**% |
| tsp225 | 14.127% | 23.313% | **11.544**% | 10.105% | 13.452% | **7.579**% |
| a280 | 20.685% | 32.413% | **14.464**% | 14.112% | 18.635% | **11.46**% |
| pcb442 | 21.608% | 31.618% | **17.406**% | 15.15% | 19.807% | **13.254**% |
| Avg. Gap | 5.953% | 13.145% | **4.55**% | 3.975% | 6.277% | **3.177**% |

In the results presented for CVRPLib in Table 9, our approach demonstrates a clear advantage over the other methods, with an average optimality gap of 3.94% in the small budget scenario and 3.344% in the median budget scenario. These results are superior to those achieved by STLbal. (6.3% and 4.63%, respectively) and UW (5.895% and 5.105%, respectively). Such findings highlight the efficacy of our method, especially under more constrained budget conditions. Furthermore, analogous to the results on TSPLib, our method exhibits superior performance particularly for instances where the problem size exceeds 100. Notably, for instances in the "X" series, our method consistently achieves the lowest optimality gaps compared to both $STL_{bal.}$ and UW. Furthermore, we present a more comprehensive comparison of results on CVRPLib X-set instances in Table 10. Our method achieves the best performance in terms of Avg. Gap under both small and median budgets, with 9.21% and 9.174% respectively. Notably, for large-scale instances (n>536) under the small budget, our method consistently outperforms all alternatives. However, it is evident that the performance improvement from small to median budget is not significant, indicating that out-of-distribution generalization to untrained scales remains a bottleneck.

## A.11    Demonstration of the Bandit Algorithms

This section presents detailed information on various bandit algorithms, as shown in Figure 9, including the selection count and average return for each task. It is evident that TS algorithm dominates in all 12 tasks, leading to poor performance on tasks where training is limited. In contrast, other bandit algorithms maintain balance across all tasks, resulting in better average results.

Table 9: Comparison results on CVRPLib. The reported results depict the optimality gap (%) ($\downarrow$) in the main aspects.

| Instances | Small Budget | | | Median Budget | | |
|---|---|---|---|---|---|---|
| | STL$_{bal.}$ | UW | Ours | STL$_{bal.}$ | UW | Ours |
| E-n76-k10 | 3.718% | 2.96% | **2.291%** | 2.78% | 2.493% | **2.49%** |
| E-n76-k7 | 5.742% | 4.817% | **2.777%** | 3.393% | 4.489% | **2.7%** |
| E-n51-k5 | **3.833%** | 6.296% | 4.167% | 5.117% | 5.183% | **3.263%** |
| E-n101-k14 | 5.439% | 6.803% | **4.439%** | 5.398% | 5.018% | **4.417%** |
| E-n22-k4 | **0.075%** | **0.075%** | **0.075%** | **0.075%** | 0.402% | **0.075%** |
| E-n101-k8 | 7.085% | 6.824% | **5.182%** | 5.109% | 5.855% | **4.247%** |
| E-n33-k4 | **1.088%** | 2.321% | 2.021% | 1.234% | 1.662% | **0.562%** |
| E-n23-k3 | 0.9% | 0.991% | **0.402%** | 0.634% | 0.634% | **0.61%** |
| E-n76-k8 | 4.534% | 3.425% | **2.486%** | 2.814% | 2.802% | **2.044%** |
| E-n76-k14 | 2.931% | 3.233% | **2.287%** | 3.571% | 3.945% | **1.767%** |
| B-n57-k9 | **2.354%** | 3.336% | 2.666% | 3.08% | 2.693% | **2.602%** |
| B-n50-k7 | 4.209% | 3.241% | **2.666%** | 2.387% | 2.387% | **2.188%** |
| B-n45-k5 | 3.276% | 3.424% | **1.554%** | 2.706% | 3.415% | **2.547%** |
| B-n64-k9 | **5.576%** | 7.892% | 6.274% | 6.225% | 5.674% | **5.527%** |
| B-n52-k7 | 3.808% | 5.024% | **2.101%** | 1.8% | 3.048% | 1.878% |
| B-n38-k6 | 2.702% | 2.983% | **1.901%** | 1.87% | 2.82% | **1.727%** |
| B-n41-k6 | 1.546% | 1.67% | **1.132%** | **0.719%** | 1.807% | 1.413% |
| B-n39-k5 | 3.877% | 2.198% | **1.623%** | 1.735% | 2.463% | **1.02%** |
| B-n63-k10 | 5.197% | 5.137% | **3.69%** | 4.31% | 5.508% | **2.268%** |
| B-n78-k10 | 7.877% | 5.947% | **5.123%** | 4.805% | 4.889% | **4.753%** |
| B-n66-k9 | 2.698% | 3.774% | **2.327%** | 3.547% | 2.984% | **1.902%** |
| B-n57-k7 | 2.612% | 1.715% | **0.434%** | 1.402% | 2.024% | **0.061%** |
| B-n45-k6 | 6.94% | 7.247% | **6.786%** | 2.885% | 6.918% | **2.257%** |
| B-n56-k7 | 6.94% | 6.823% | **4.134%** | 4.106% | 5.888% | **2.848%** |
| B-n67-k10 | 4.882% | 4.788% | **4.574%** | 4.793% | 4.582% | **3.959%** |
| B-n34-k5 | 2.409% | 1.404% | **1.278%** | 1.201% | 1.911% | **1.156%** |
| Continued in next column | | | | | | |

| | STL$_{bal.}$ | UW | Ours | STL$_{bal.}$ | UW | Ours |
|---|---|---|---|---|---|---|
| B-n35-k5 | 2.804% | 2.746% | **1.605%** | 1.915% | 3.375% | **1.449%** |
| B-n31-k5 | 2.535% | **2.164%** | 2.201% | **0.887%** | 2.868% | 1.734% |
| B-n43-k6 | 2.07% | 1.612% | **1.328%** | 1.433% | 1.162% | **1.091%** |
| B-n50-k8 | 2.653% | 2.22% | **0.753%** | 2.081% | 1.537% | **0.908%** |
| B-n44-k7 | 4.102% | 4.496% | **2.999%** | **1.648%** | 4.506% | 2.935% |
| B-n68-k9 | 2.978% | 3.814% | 3.087% | **2.005%** | 3.628% | 2.674% |
| X-n129-k18 | 4.634% | 4.969% | **3.272%** | 4.152% | 4.443% | **1.606%** |
| X-n157-k13 | 13.281% | 7.763% | **3.434%** | 8.206% | 6.047% | **3.243%** |
| X-n162-k11 | 9.169% | 7.895% | **4.925%** | 5.887% | 6.812% | **3.77%** |
| X-n106-k14 | 6.514% | 5.357% | **4.021%** | 5.848% | 4.832% | **2.897%** |
| X-n153-k22 | 14.706% | 12.736% | **9.842%** | 13.081% | 12.746% | **11.582%** |
| X-n172-k51 | 9.696% | 8.034% | **5.79%** | 8.685% | 7.601% | **5.574%** |
| X-n143-k7 | 13.044% | 11.111% | **5.389%** | 8.501% | 9.398% | **5.679%** |
| X-n139-k10 | 6.853% | 6.412% | **3.201%** | 4.548% | 4.61% | **2.667%** |
| X-n167-k10 | 9.207% | 9.577% | **5.261%** | 6.896% | 7.714% | **4.51%** |
| X-n176-k26 | 9.509% | 9.657% | **8.049%** | 9.96% | **9.42%** | 10.584% |
| X-n134-k13 | 12.317% | 11.499% | **6.519%** | 9.79% | 9.204% | **5.727%** |
| X-n125-k30 | 6.379% | **5.126%** | 5.176% | 5.327% | 5.31% | **4.863%** |
| X-n181-k23 | 7.51% | 5.866% | **3.14%** | 4.759% | 4.317% | **2.812%** |
| X-n101-k25 | 7.612% | 6.11% | **4.17%** | 6.356% | 4.849% | **4.715%** |
| X-n120-k6 | 11.344% | 9.381% | **5.698%** | 6.232% | 7.243% | **4.003%** |
| X-n110-k13 | 3.906% | 5.189% | **3.022%** | 2.966% | 3.243% | **1.866%** |
| X-n115-k10 | 7.958% | 8.94% | **4.115%** | 5.348% | 4.232% | **2.674%** |
| X-n148-k46 | 9.003% | 7.867% | **5.387%** | 7.881% | 6.486% | **5.012%** |
| F-n45-k4 | 4.241% | 4.458% | **3.135%** | 3.875% | 4.323% | **2.144%** |
| F-n135-k7 | 30.986% | 28.722% | **16.019%** | 16.893% | 24.714% | **8.727%** |
| F-n72-k4 | 16.616% | 14.387% | **12.906%** | 12.541% | 14.458% | **11.514%** |
| Avg. Gap | 6.3% | 5.895% | **3.94%** | 4.63% | 5.105% | **3.344%** |

## A.12 Stability of Each Method

This section examines the stability characteristics of the compared methods presented in Figure 11. The confidence interval plots reveal significant differences in stability among the various approaches. For small-scale problems such as TSP20 and CVRP20, most methods demonstrate relatively narrow confidence intervals, indicating consistent performance across trials. However, approaches like CAGrad and Nash-MTL exhibit increasingly wide error bars on large-scale problems, suggesting high variance in solution quality. In contrast, our proposed method maintains remarkably tight confidence intervals across all problem scales and budget configurations. For instance, in the TSP100 scenario, while methods like TAG and UW show error bars spanning over 2% in optimality gap, our approach maintains a confidence interval below 1%, demonstrating exceptional stability. Similarly, for CVRP100, our method achieves not only the lowest mean optimality gap but also the narrowest confidence interval among all compared approaches, particularly notable given the inherent complexity of vehicle routing problems. This stability advantage extends to OP100 and KP200, where our method consistently delivers reliable performance regardless of the computational budget allocated.

We further demonstrate the stability of each model obtained by the specific methods on 10000 test instances from each COP and the corresponding 2-sigma error bar plot is shown in Figure 10. It is generally accepted that longer training epochs lead to reduced standard variance for each method. Additionally, our method produces a model with the most stable performance in most scenarios when compared to other MTL methods across almost all cases.

## A.13 Additional Experiments on Other Domains

We select the challenge domain on Time Series to evaluate the performance of our method. Following the common practice in this domain, there are multiple series in one piece of data and the prediction on each series is seen as a task. [2]We consider Long-term Forecasting tasks comprising ETT (4 subsets), Weather,

---

[2]For forecasting and imputation tasks, we ignore the Electricity and Traffic dataset because all MTL methods meet out of memory errors because there are too many tasks.

Table 10: Comparison results on CVRPLib large X-set instances. The reported results depict the optimality gap (%) (↓) in the main aspects.

| Instances | Small Budget | | | Median Budget | | |
|---|---|---|---|---|---|---|
| | STL$_{bal.}$ | UW | Ours | STL$_{bal.}$ | UW | Ours |
| X-n153-k22 | 14.706% | 12.736% | **9.842%** | 13.081% | 12.746% | 11.582% |
| X-n157-k13 | 13.281% | 7.763% | **3.434%** | 8.206% | 6.047% | 3.243% |
| X-n162-k11 | 9.169% | 7.895% | **4.925%** | 5.887% | 6.812% | 3.77% |
| X-n167-k10 | 9.207% | 9.577% | **5.261%** | 6.896% | 7.714% | 4.51% |
| X-n172-k51 | 9.696% | 8.034% | **5.79%** | 8.685% | 7.601% | 5.574% |
| X-n176-k26 | 9.509% | 9.657% | **8.049%** | 9.96% | **9.42%** | 10.584% |
| X-n181-k23 | 7.51% | 5.866% | **3.14%** | 4.759% | 4.317% | **2.812%** |
| X-n186-k15 | 8.134% | 6.2% | **4.538%** | 5.593% | 5.593% | **3.55%** |
| X-n190-k8 | 23.249% | 22.66% | **11.446%** | 15.895% | 19.829% | **8.168%** |
| X-n195-k51 | 11.579% | 8.789% | **6.58%** | 8.67% | 8.08% | 7.032% |
| X-n200-k36 | 7.974% | 9.087% | **5.255%** | **6.128%** | 6.436% | 6.136% |
| X-n204-k19 | 10.312% | 7.49% | **4.979%** | 7.239% | 6.413% | 4.065% |
| X-n209-k16 | 9.162% | 8.672% | **5.258%** | 6.981% | 7.342% | 4.02% |
| X-n214-k11 | 20.68% | 17.096% | **8.92%** | 15.749% | 13.682% | 7.321% |
| X-n219-k73 | 4.375% | **3.497%** | 4.571% | 3.615% | **3.233%** | 9.83% |
| X-n223-k34 | 7.853% | 6.587% | **4.546%** | 7.276% | 5.984% | 4.405% |
| X-n228-k23 | 14.332% | 13.957% | **7.996%** | 12.66% | 14.121% | 9.17% |
| X-n233-k16 | 15.036% | 13.044% | **7.36%** | 11.054% | 10.225% | 6.318% |
| X-n237-k14 | 12.34% | 9.22% | **5.873%** | 9.229% | 8.42% | 4.985% |
| X-n242-k48 | 7.898% | 6.473% | **4.18%** | 5.885% | 5.295% | 3.768% |
| X-n247-k50 | 13.286% | 11.53% | **9.837%** | 12.483% | **11.252%** | 12.42% |
| X-n251-k28 | 9.639% | 6.9% | **5.38%** | 8.354% | 7.026% | 4.555% |
| X-n256-k16 | 12.559% | 11.406% | **7.483%** | 10.57% | 9.952% | 5.664% |
| X-n261-k13 | 15.276% | 12.801% | **8.105%** | 11.256% | 11.87% | 6.037% |
| X-n266-k58 | 11.201% | 9.328% | **6.785%** | 9.059% | 7.605% | 7.123% |
| X-n270-k35 | 10.776% | 9.511% | **5.699%** | 6.707% | 6.614% | 4.996% |
| X-n275-k28 | 12.332% | 9.559% | **7.728%** | 11.486% | **7.87%** | 11.82% |
| X-n280-k17 | 14.627% | 12.623% | **8.265%** | 9.79% | 12.037% | 7.689% |
| X-n284-k15 | 28.634% | 19.965% | **11.425%** | 17.151% | 17.287% | 9.917% |
| X-n289-k60 | 10.273% | 9.286% | **6.553%** | 8.843% | 7.496% | 6.114% |
| X-n294-k50 | 11.923% | 8.741% | **6.137%** | 10.047% | 7.505% | 6.46% |
| X-n298-k31 | 11.893% | 10.425% | **6.204%** | 9.784% | 9.086% | 5.791% |
| X-n303-k21 | 15.677% | 12.521% | **7.1%** | 12.006% | 10.066% | 6.113% |
| X-n308-k13 | 20.819% | 17.586% | **11.602%** | 13.931% | 15.671% | 8.787% |
| X-n313-k71 | 10.101% | 7.316% | **6.385%** | 8.206% | 7.284% | 5.154% |
| X-n317-k53 | 8.573% | 7.698% | **6.984%** | 7.461% | **5.571%** | 7.512% |
| X-n322-k28 | 13.077% | 9.637% | **7.348%** | 10.97% | 8.948% | 6.152% |
| X-n327-k20 | 16.5% | 12.753% | **8.583%** | 13.25% | 11.878% | 7.405% |
| X-n331-k15 | 17.917% | 14.3% | **8.582%** | 12.255% | 12.554% | 7.65% |
| X-n336-k84 | 10.591% | 8.832% | **6.135%** | 8.698% | 6.894% | 5.45% |
| X-n344-k43 | 12.481% | 9.243% | **6.333%** | 9.793% | 8.089% | 5.927% |
| X-n351-k40 | 23.583% | 19.07% | **10.175%** | 19.736% | 13.74% | 8.873% |
| X-n359-k29 | 13.357% | 10.196% | **6.311%** | 8.97% | 9.258% | **5.188%** |
| X-n367-k17 | 26.92% | 25.243% | **12.559%** | 22.251% | 17.674% | **11.841%** |
| X-n376-k94 | 8.37% | 5.268% | **4.461%** | 6.62% | **4.62%** | 19.442% |
| X-n384-k52 | 12.47% | 9.467% | **7.249%** | 8.922% | 8.793% | **5.539%** |
| X-n393-k38 | 15.021% | 11.46% | **7.761%** | 13.005% | 10.953% | **7.308%** |
| X-n401-k29 | 14.462% | 12.664% | **6.402%** | 10.178% | 12.125% | **5.985%** |
| X-n411-k19 | 31.257% | 29.84% | **17.378%** | 26.846% | 22.044% | **14.923%** |
| X-n420-k130 | 13.386% | 9.999% | **7.311%** | 12.665% | 9.314% | **7.341%** |
| X-n429-k61 | 14.223% | 10.818% | **7.665%** | 10.63% | 8.634% | **7.716%** |
| X-n439-k37 | 17.993% | **10.405%** | 11.945% | 14.041% | **9.48%** | 19.02% |
| X-n449-k29 | 17.373% | 12.241% | **8.217%** | 12.251% | 11.785% | **7.431%** |
| X-n459-k26 | 35.717% | 33.254% | **15.816%** | 27.829% | 25.709% | **13.068%** |
| X-n469-k138 | 12.839% | 10.714% | **8.997%** | 11.404% | **9.554%** | 13.673% |
| X-n480-k70 | 13.093% | 10.31% | **7.602%** | 9.71% | **8.144%** | 8.86% |
| X-n491-k59 | 12.518% | 11.1% | **7.449%** | 11.76% | 8.725% | **7.053%** |
| X-n502-k39 | 31.767% | **11.941%** | 16.177% | 13.729% | 14.196% | **13.057%** |
| X-n513-k21 | 36.662% | 22.441% | **14.372%** | 28.895% | 21.226% | **12.102%** |
| X-n524-k153 | 11.975% | **10.539%** | 10.766% | 12.89% | **10.372%** | 11.341% |
| X-n536-k96 | 15.463% | 13.286% | **9.865%** | 14.05% | 12.02% | **9.003%** |
| X-n548-k50 | 13.354% | 10.535% | **6.862%** | 10.33% | **8.727%** | 9.076% |
| X-n561-k42 | 21.571% | 13.463% | **10.129%** | 16.558% | 11.263% | **8.809%** |
| X-n573-k30 | 57.107% | 37.495% | **21.593%** | 25.765% | 29.072% | **15.744%** |
| X-n586-k159 | 14.7% | 12.448% | **9.937%** | 12.221% | **10.015%** | 12.783% |
| X-n599-k92 | 15.264% | 12.339% | **8.936%** | 11.369% | **10.018%** | 10.044% |
| X-n613-k62 | 18.895% | 14.054% | **8.76%** | 15.746% | 11.519% | **9.232%** |
| X-n627-k43 | 32.829% | 29.237% | **15.841%** | 21.009% | 17.086% | **11.597%** |
| X-n641-k35 | 21.398% | 16.795% | **11.724%** | 15.384% | 14.725% | **10.383%** |
| X-n670-k130 | 18.374% | 14.581% | **12.882%** | 17.64% | **13.113%** | 14.493% |
| X-n685-k75 | 22.086% | 14.013% | **10.585%** | 16.578% | 12.948% | **10.125%** |
| X-n701-k44 | 20.035% | 13.744% | **9.216%** | 13.87% | 14.069% | **8.404%** |
| X-n716-k35 | 52.99% | 27.968% | **17.734%** | 31.692% | 25.924% | **15.159%** |
| X-n733-k159 | 18.048% | 12.284% | **9.352%** | 13.937% | 9.87% | **9.005%** |
| X-n749-k98 | 24.317% | 17.183% | **12.182%** | 19.278% | 14.994% | **10.94%** |
| X-n766-k71 | 19.917% | 16.109% | **11.691%** | 14.925% | 14.048% | **10.732%** |
| X-n783-k48 | 26.122% | 17.523% | **12.083%** | 17.502% | 15.687% | **10.984%** |
| X-n801-k40 | 25.352% | 16.169% | **13.368%** | 18.48% | **16.591%** | 17.866% |
| X-n837-k142 | 16.701% | 11.45% | **8.473%** | 12.523% | **10.398%** | 10.42% |
| X-n876-k59 | 32.379% | 26.154% | **14.322%** | 16.465% | 20.461% | **11.984%** |
| X-n895-k37 | 35.09% | 22.169% | **15.525%** | 23.595% | 21.286% | **16.001%** |
| X-n916-k207 | 15.837% | 14.357% | **11.17%** | 10.88% | **10.047%** | 12.44% |
| X-n936-k151 | 24.645% | 18.922% | **17.4%** | 23.302% | **17.602%** | 20.523% |
| X-n957-k87 | 28.359% | 17.266% | **13.175%** | 21.262% | **13.764%** | 16.707% |
| X-n979-k58 | 21.886% | 28.181% | **13.832%** | 14.21% | 18.18% | **13.084%** |
| X-n1001-k43 | 33.074% | 23.709% | **16.201%** | 23.08% | 20.05% | **14.098%** |
| Avg. Gap | 17.709% | 13.659% | **9.21%** | 13.134% | 11.655% | **9.174%** |

Continued in next column

Table 11: For Long-term Forecasting tasks, all the results are averaged from 4 different prediction lengths, that is {24, 36, 48, 60} for ILI and {96, 192, 336, 720} for the others. "Baseline" provides the the results in the original paper. Results in bold mean achieving the best performance among all methods.

| Method | ETT-h1 | | ETT-h2 | | ETT-m1 | | ETT-m2 | | Whether | | Exchange | | ILI | |
|---|---|---|---|---|---|---|---|---|---|---|---|---|---|---|
| | MSE | MAE | MSE | MAE | MSE | MAE | MSE | MAE | MSE | MAE | MSE | MAE | MSE | MAE |
| MTL | 0.496 | 0.487 | 0.450 | 0.459 | 0.588 | 0.517 | 0.327 | 0.371 | 0.338 | 0.382 | 0.613 | 0.539 | 3.006 | **1.161** |
| Bandit-MTL | 0.438 | 0.420 | 0.398 | 0.363 | 0.533 | 0.643 | 0.304 | 0.220 | 0.327 | 0.254 | 0.319 | 0.181 | 1.424 | 3.955 |
| UW | 0.420 | 0.385 | 0.400 | 0.359 | 0.502 | 0.557 | 0.325 | 0.236 | **0.308** | **0.231** | 0.287 | 0.153 | 1.425 | 3.942 |
| CAGrad | 0.468 | 0.466 | 0.390 | 0.351 | 0.477 | 0.488 | 0.304 | 0.220 | 0.312 | 0.241 | 0.310 | 0.174 | 1.411 | 3.856 |
| IMTL-G | 0.445 | 0.423 | 0.392 | 0.352 | **0.465** | **0.462** | 0.303 | 0.219 | 0.319 | 0.245 | **0.286** | **0.153** | 1.399 | 3.776 |
| Nash-MTL | 0.468 | 0.472 | 0.409 | 0.370 | 0.468 | 0.483 | 0.310 | 0.225 | 0.315 | 0.240 | 0.302 | 0.165 | **1.376** | 3.806 |
| Ours | **0.418** | **0.385** | **0.383** | **0.343** | 0.506 | 0.547 | **0.299** | **0.215** | 0.360 | 0.277 | 0.333 | 0.193 | 1.689 | 5.189 |

Exchange and ILI datasets, and Imputation task comprising ETT and Weather. The backbone is AutoFormer (Wu et al., 2021a) and all the experimental settings keep the same as the original paper.

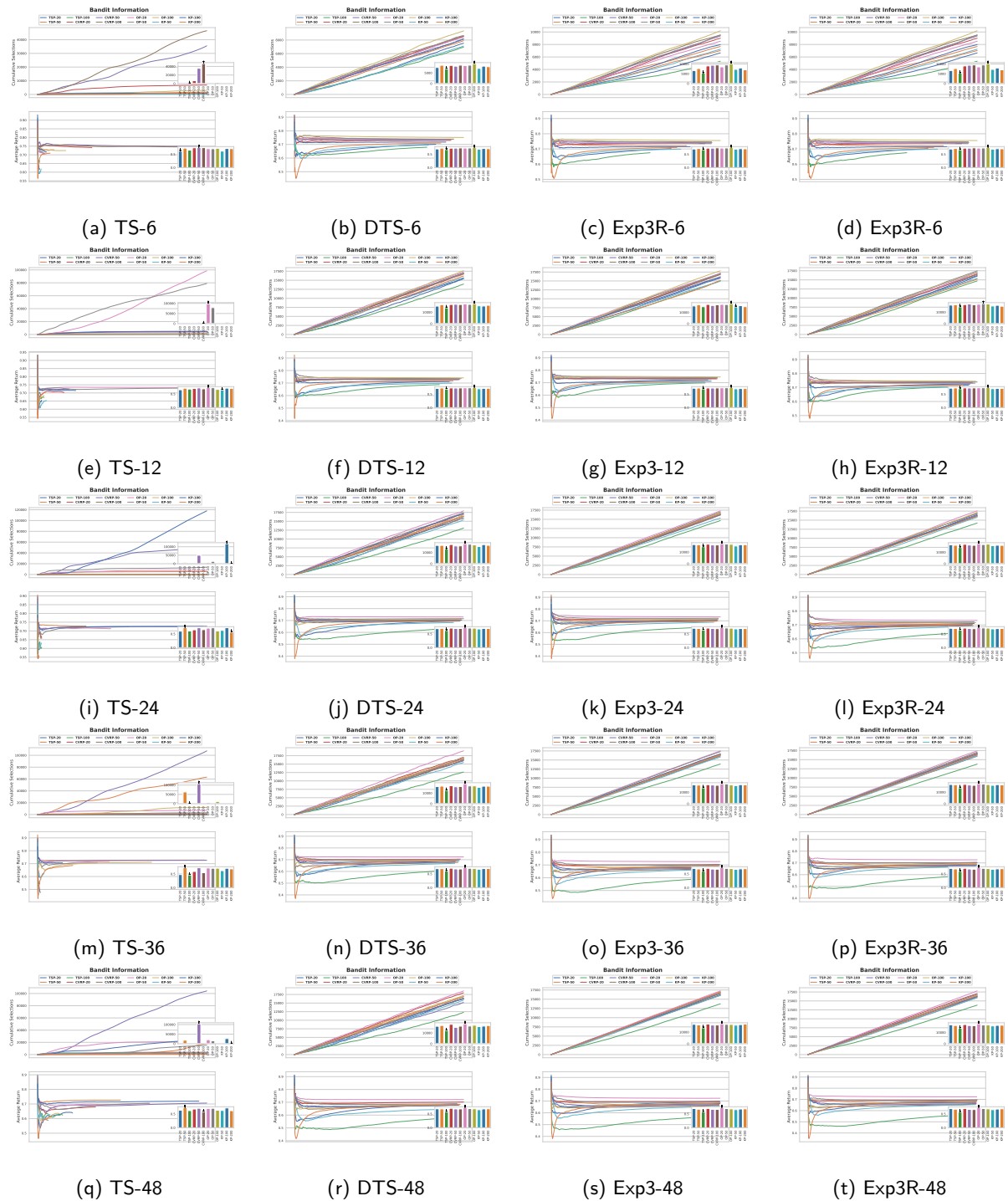

(a) TS-6     (b) DTS-6     (c) Exp3R-6     (d) Exp3R-6

(e) TS-12     (f) DTS-12     (g) Exp3-12     (h) Exp3R-12

(i) TS-24     (j) DTS-24     (k) Exp3-24     (l) Exp3R-24

(m) TS-36     (n) DTS-36     (o) Exp3-36     (p) Exp3R-36

(q) TS-48     (r) DTS-48     (s) Exp3-48     (t) Exp3R-48

Figure 9: Further results of the bandit information. The caption of each subfigure "A-B" means the influence matrix obtained by algorithm A with update frequency B.

Results show that there are no consisting best methods for all datasets, however, our method can achieve the best performance consistently on 3 out of 7 datasets.

From these results, our method performs well in some cases, but generally speaking, there is no one universal approach which can handle all tasks or even on all datasets in a task.

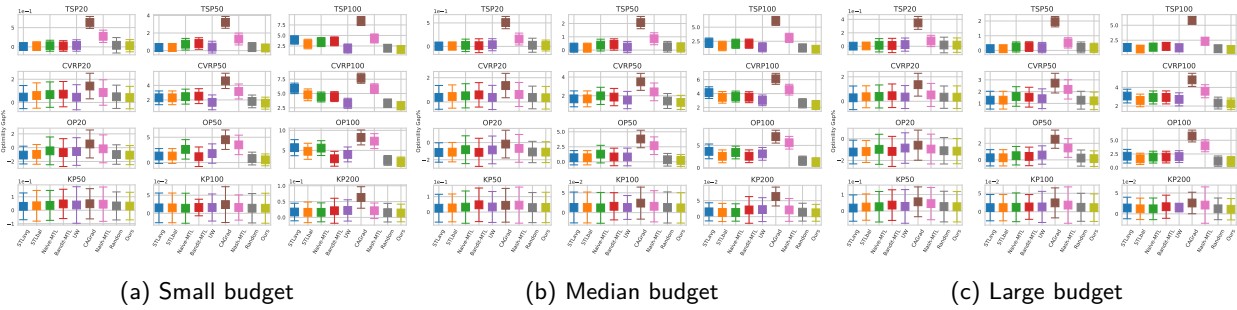

(a) Small budget      (b) Median budget      (c) Large budget

Figure 10: Stability of the model obtained by different methods on 10000 instances from each COP with different budget allocations.

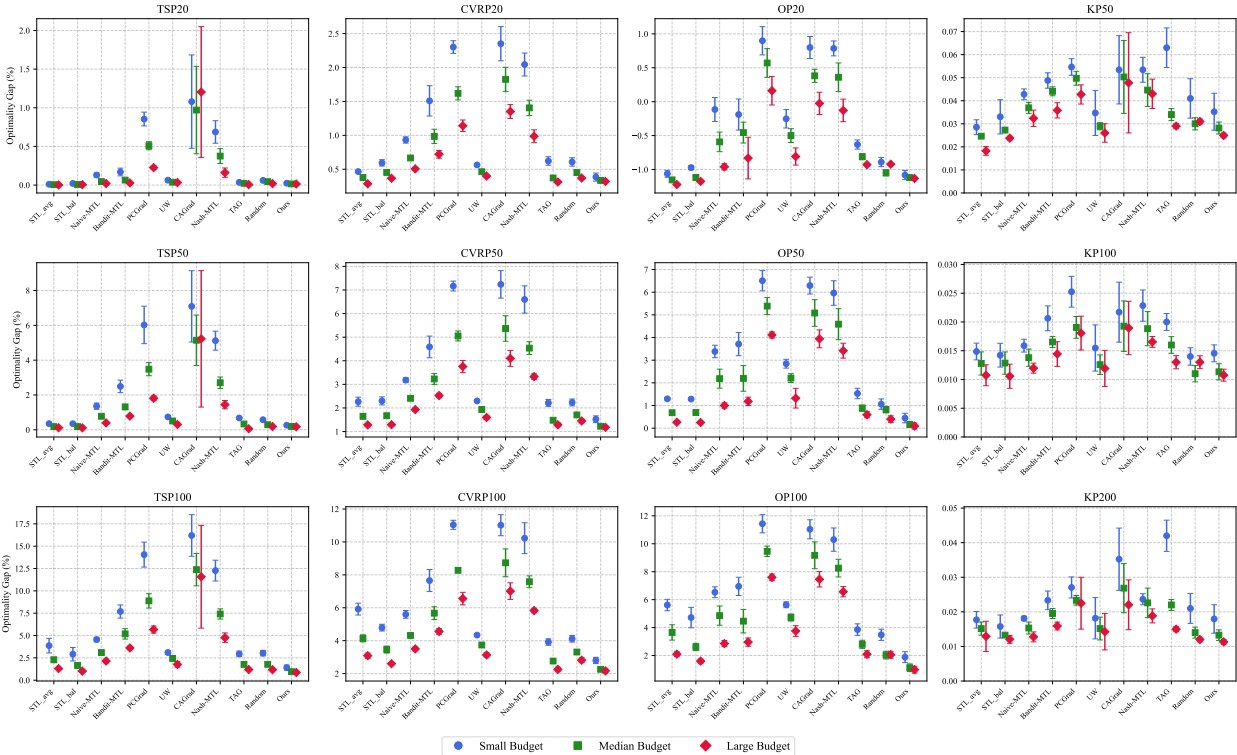

Figure 11: Comparison of methods across different budgets (small, median, and large) on TSP, CVRP, OP, and KP with varying problem scales. The results are presented as the optimality gap (%) for each method, with error bars indicating the standard deviation over 5 runs.

## Broader Impact Statement

Our contributions significantly impact machine learning and its applications by introducing a novel framework for training combinatorial neural solvers with multi-task learning. Based on this, we not only enhance performance over standard paradigms but also offer a scalable method for training large models under resource constraints. This advancement has the potential to revolutionize industries that rely on complex optimization, such as logistics and manufacturing, by enabling more sophisticated AI solutions. Furthermore, our theoretical insights into loss decomposition and the introduction of an influence matrix shed light on the inner workings of neural solvers, facilitating a deeper understanding and improvement of machine learning models. However, the deployment of these advanced technologies also necessitates a careful exami-

Table 12: For Imputation tasks, time series are randomly masked $\{12.5\%, 25\%, 37.5\%, 50\%\}$ time points in length-96. The results are averaged from 4 different mask ratios. Results with underline mean achieving the best performance among all methods and those in bold mean achieving the best among all MTL methods.

| Method | ETT-h1 MSE | ETT-h1 MAE | ETT-h2 MSE | ETT-h2 MAE | ETT-m1 MSE | ETT-m1 MAE | ETT-m2 MSE | ETT-m2 MAE | Weather MSE | Weather MAE |
|---|---|---|---|---|---|---|---|---|---|---|
| Baseline | 0.103 | 0.214 | 0.055 | 0.156 | 0.051 | 0.150 | 0.029 | 0.105 | 0.031 | 0.057 |
| Bandit-MTL | 0.324 | 0.201 | 0.437 | 0.414 | 0.579 | 0.576 | **0.603** | 0.792 | **0.255** | **0.154** |
| UW | **0.268** | **0.143** | 0.354 | 0.280 | 0.669 | 0.767 | 0.774 | 1.124 | 0.391 | 0.324 |
| CAGrad | 0.269 | 0.144 | 0.354 | 0.267 | 0.593 | 0.605 | 0.640 | 0.747 | 0.347 | 0.257 |
| IMTL | 0.270 | 0.145 | 0.353 | 0.266 | 0.594 | 0.605 | 0.681 | 0.854 | 0.388 | 0.337 |
| Nash-MTL | 0.268 | 0.143 | **0.347** | **0.255** | 0.637 | 0.694 | 0.693 | 0.866 | 0.489 | 0.494 |
| Ours | 0.300 | 0.174 | 0.411 | 0.366 | **0.473** | **0.384** | 0.609 | **0.734** | 0.283 | 0.174 |

nation of ethical and societal implications, including potential impacts on employment and the importance of ensuring privacy, security, and fairness in AI applications.

