# OpenReview forum: "Efficient Training of Multi-task Neural Solver for Combinatorial Optimization"
_TMLR — Accepted by TMLR_

### Review · Reviewer_ZdYQ · 2025-02-10

**Summary Of Contributions:**

The paper presents a framework for efficiently training a multi-task neural solver for combinatorial optimization problems (COPs). The authors propose a multi-armed bandit (MAB) approach to dynamically select tasks during training, which avoids the need for explicit loss balancing in traditional multi-task learning (MTL). They also provide theoretical loss decomposition under an encoder-decoder framework, leading to the construction of an influence matrix that reveals task relationships. Empirical validation of the approach on sampled instances and benchmark datasets (TSPLib and CVRPLib), demonstrating superior performance compared to many single-task learning (STL) and MTL baselines.

**Audience:**

Yes

**Claims And Evidence:**

Yes

**Requested Changes:**

1. The influence matrix is a key component of the proposed method. While the authors argue that it captures task relationships, it would be beneficial to see a more detailed analysis of how this matrix evolves during training and how it impacts the selection of tasks. Moreover, Figure 4 shows that there are no (or little) connections between different problem types. In this case, why should we train them together with a shared encoder? Why not train one model for each problem type?

2. The paper compares the proposed method with several MTL baselines and STL. However, the comparison with STL seems to be based on different training budgets. Could the authors clarify if the comparison is fair in terms of computational resources and training time? Morevoer, when the training budget of STL increases, will the final converged results outperform MTL methods including the one proposed in this paper?

3. The paper focuses on combinatorial optimization problems. How well does the proposed method generalize to other domains, such as natural language processing or computer vision? Are there any limitations or specific conditions (such as strong or weak relationships) under which the method performs well?

4. The performance of the proposed method may depend on the choice of hyperparameters, such as the update frequency of the bandit algorithm. Can the authors provide more analysis to show how robust the method is to different hyperparameter settings?

5. While the paper demonstrates results on TSPLib and CVRPLib, the instances selected are of small sizes. It would be valuable to see how the method performs on large-scale instances, such as CVRPLib X-set with up to 1,000 nodes. Will the MTL benefit large-scale generalization when compared to STL on CVRP?

**Strengths And Weaknesses:**

Strengths:
1. The proposed method is general and efficient.
2. The comparison to baselines is comprehensive.

Weaknesses:
More experimental results and additional clarifications are suggested to support the authors' claims.

---

> ### Author Response · Authors · 2025-02-13
> **Response Part 1/2**
>
> We sincerely thank the reviewer for the thoughtful comments and valuable suggestions. Given the word limit, we will address the responses in two parts. Below are our first set of point-by-point responses.
> 1. -  In Section 3.2, we describe how the influence matrix guides task selection through the design of the bandit algorithm's reward function (Equation 9). Regarding the evolution of the influence matrix during training, directly visualizing this $n\times n$ matrix (where n is the number of tasks) presents significant challenges. Instead, we provide an alternative analysis in Appendix 10 (Figure 9), which shows the cumulative selection count of each task and their corresponding average returns throughout the training process. These metrics indirectly reflect the evolution of the influence matrix.
> As demonstrated in Figure 9, Thompson Sampling tends to lead to task domination, where certain tasks are selected disproportionately, potentially undermining the training of other tasks. In contrast, other bandit algorithms (Discounted Thompson Sampling, EXP3, and EXP3.R) maintain a more balanced task selection strategy. A detailed analysis of these behavioral patterns can be found in Appendix 10.
>     - Our decision to use a shared encoder is based on several key considerations:
>         - First, we assume potential positive transfer exists between different COP types, and a shared encoder serves as a bridge to facilitate this transfer - a common practice in multi-task learning. Using separate encoders would effectively isolate the training of different problem types, hindering potential positive transfer between them.
>          - Second, the limited connections observed between different problem types are actually findings that emerged after using the shared encoder. Rather than using these findings to justify separate encoders, they motivate us to design better neural solver backbones that can enhance transfer between different COP types. However, designing neural solver backbones for cross-problem-type transfer is a novel topic that has not been explored in existing research and is beyond the scope of our current work. We discuss this promising direction in the Future Work section.
>         - Third, using separate encoders would introduce significant practical challenges. Since the majority of model parameters are concentrated in the encoder, separate encoders would substantially increase both training costs and storage complexity. Moreover, this approach would compromise the generality of our method - without prior knowledge of task relationships, determining how to group tasks for separate encoders becomes a critical challenge that can be avoided by using a shared encoder.
>
> 2. We appreciate your bringing up this important point about training budgets. Let us clarify the comparison details:
>     - For the results in Table 2, all baselines were evaluated under identical training budgets (in GPU hours).
>     - The results in Table 4 were indeed obtained under similar training epochs (1000 vs. 1200, with 100 epochs per task) rather than identical budgets. However, when we examine the actual training time using 8 A100 GPUs, the STL approach required 3.48 days compared to our method's 3.28 days. Notably, even in this comparison that favors STL, our method still demonstrates superior performance.
>     - In Figure 3, we deliberately extended this comparison to showcase our method's generalization capabilities. We compared a single model trained using our approach for 1000 epochs against 12 separate task-specific models, each trained for 1000 epochs (effectively comparing 1000 epochs vs. 12,000 epochs, or 3.28 days vs. 34.8 days in terms of training budget). Despite this substantial resource advantage for the STL approach, our method exhibited better generalization across different tasks, as detailed in Section 4.2.
>
>     For the question:
>
>     > when the training budget of STL increases, will the final converged results outperform MTL methods?
>
>     we can examine this through the generalization results presented in Figure 3. As training epochs increase, while STL performance tends to stabilize, our method with median budgets (3.28 days of GPU hours) outperforms STL across all problem types except TSP. For TSP, our model achieves performance comparable to STL at around 500 epochs. Notably, training these STL models requires 34.8 days, which is substantially higher than the median budget (3.28 days) used in our approach. These results demonstrate that even with increased training budgets, STL fails to consistently outperform our method, despite consuming significantly more computational resources.

---

> > ### Author Response · Authors · 2025-02-13
> > **Response Part 2/2**
> >
> > 3. We appreciate your question about generalization to NLP and CV domains. While investigating these domains is beyond the scope of our current work focusing on COPs, we did explore generalization to the time series domain as reported in Appendix A.12. The results show varying performance across different tasks: while our method achieves the best results across multiple datasets in Long-term Forecasting Tasks (Table 10), for Imputation Tasks, all MTL methods, including ours, fail to outperform the naive MTL baseline. These results suggest that the effectiveness of MTL methods can be significantly influenced by task characteristics and dataset properties. While we cannot claim universal superiority, our work demonstrates consistent and superior performance, specifically in the domain of COPs.
> >
> > 4. We address the influence of different bandit algorithms and their update frequencies in Appendix A.4. As shown in Figure 6, while Thompson Sampling exhibits sensitivity to the update frequency, the other three bandit algorithms (Discounted Thompson Sampling, EXP3, and EXP3.R) demonstrate robust performance across different frequency settings. A detailed analysis of these results can be found in Appendix A.4.
> >
> > 5. Thank you for this suggestion. We have conducted additional experiments on CVRP X-set instances, with problem sizes ranging from 100 to 1,000 nodes. The results further validate our method's effectiveness on large-scale instances in terms of Avg. Gap:
> >
> > |             | STL-small   | UW-small         | Ours-small        | STL-median        | UW-median        | Ours-median       |
> > |:------------|:------------|:------------------|:------------------|:------------------|:------------------|:------------------|
> > | Avg. Gap    | 16.624%    | 12.955%          | **8.681%**  | 12.335%           | 10.991%          | **8.564%** |
> >
> > Specifically, for the results of large-scale instances (700-1000 nodes) shown below,
> >
> > |             | STL-small   | UW-small         | Ours-small        | STL-median        | UW-median        | Ours-median       |
> > |:------------|:------------|:------------------|:------------------|:------------------|:------------------|:------------------|
> > | X-n701-k44  | 20.035%     | 13.744%           | **9.216%**        | 13.87%            | 14.069%           | **8.404%**        |
> > | X-n716-k35  | 52.99%      | 27.968%           | **17.734%**       | 31.692%           | 25.924%           | **15.159%**       |
> > | X-n733-k159 | 18.048%     | 12.284%           | **9.352%**        | 13.937%           | 9.87%             | **9.005%**        |
> > | X-n749-k98  | 24.317%     | 17.183%           | **12.182%**       | 19.278%           | 14.994%           | **10.94%**        |
> > | X-n766-k71  | 19.917%     | 16.109%           | **11.691%**       | 14.925%           | 14.048%           | **10.732%**       |
> > | X-n783-k48  | 26.122%     | 17.523%           | **12.083%**       | 17.502%           | 15.687%           | **10.984%**       |
> > | X-n801-k40  | 25.352%     | 16.169%           | **13.368%**       | 18.48%            | **16.591%**       | 17.866%           |
> > | X-n837-k142 | 16.701%     | 11.45%            | **8.473%**        | 12.523%           | **10.398%**       | 10.42%            |
> > | X-n876-k59  | 32.379%     | 26.154%           | **14.322%**       | 16.465%           | 20.461%           | **11.984%**       |
> > | X-n895-k37  | 35.09%      | 22.169%           | **15.525%**       | 23.595%           | 21.286%           | **16.001%**       |
> > | X-n916-k207 | 15.837%     | 14.357%           | **11.17%**        | 10.88%            | **10.047%**       | 12.44%            |
> > | X-n936-k151 | 24.645%     | 18.922%           | **17.4%**         | 23.302%           | **17.602%**       | 20.523%           |
> > | X-n957-k87  | 28.359%     | 17.266%           | **13.175%**       | 21.262%           | **13.764%**       | 16.707%           |
> > | X-n979-k58  | 21.886%     | 28.181%           | **13.832%**       | 14.21%            | 18.18%            | **13.084%**       |
> > | X-n1001-k43 | 33.074%     | 23.709%           | **16.201%**       | 23.08%            | 20.05%            | **14.098%**       |
> >
> > we observe consistent performance advantages on the case of small budgets.

---

> > ### Author Response · Authors · 2025-02-17
> > **Supplementary Response to Requested Change 1**
> >
> > Regarding "Why not train one model for each problem type?", your intuition is correct.
> >
> > Under the current backbone, since different COP types occupy independent subspaces, training separate models for each COP type could improve parameter utilization. We verified this in Appendix A.8: under the same experimental settings, training four groups of COPs separately (Ours-4G) outperforms training all tasks together (Ours-12T).
> >
> > However, it's crucial to emphasize that **this is a finding that emerged after applying our method**. In general practical scenarios, such grouping strategies are **not known a priori**. Additionally, this observation motivates us to design new backbones and training algorithms to enhance positive transfer between different types of tasks, thereby improving model parameter utilization.

---

### Review · Reviewer_W2Yt · 2025-02-13

**Summary Of Contributions:**

This work considers the problem of training a neural model that can solve several combinatorial optimization problems (COPs) at once. It casts this problem in a multi-task learning (MTL) context. The model architecture contains parameters that are shared between tasks as well as some that are specialized per-task. It proposes using a bandit algorithm to select, within the training loop, what should be the next COP to train on. The reward for this bandit is designed based on task losses, which the authors show are decomposable. The authors conduct experiments where they compare their method with an extensive range of MTL methods, generally showing favorable performance over them.

**Audience:**

Yes

**Broader Impact Concerns:**

None.

**Claims And Evidence:**

Yes

**Requested Changes:**

C1. Please address W1 above, which I view as a mandatory condition for securing acceptance.

C2. There is little to no detail about the workings of the other MTL methods, which are essentially the baselines for this work. I believe the paper would benefit from adding a high-level discussion (perhaps to the appendix if space is limited) of how the other methods operate. It is difficult to judge the complexity analysis in Table 1 without it. It should also be clear, in my opinion, *why* we see that their performance is not so great in this setting, and the proposed method is able to perform better.

C3. It does not seem that the proposed method is dependent on the tasks themselves being COPs? Is this indeed the case or not?

C4. Small points: many of the equation refs are of the form "Equation equation 1", consider removing the second "equation" throughout the manuscript. Typos: "combinarotial" (page 2)

**Strengths And Weaknesses:**

### Strengths

S1. In my view, the main strength of the method proposed by the authors is that it is very general and does not depend on the specifics of the combinatorial optimization problem. In fact, the "inner loop" algorithm does not even need to be based on supervised learning -- indeed, the authors use a reinforcement learning method, which does not require ground truth labels for training. The method is a framework in the true sense of the word, and it is not a stretch to imagine it being used for other COPs by swapping out the neural solvers. In my view, it is an appealing proposal for "foundation models" for COPs.

S2. The analysis around the influence of tasks on each other is very interesting -- and it is a nice property of the proposed framework that it enables such an analysis in the first place. It may also give a way of "quantifying" how close COPs are to each other.

S3. The authors consider a wide range of MTL and STL setups in their evaluation, so the evaluation is quite robust in this sense.

### Weaknesses

W1. A doubt that I have about the reported results is that there are no confidence intervals or error bars (on the random initialization of model weights) reported in the tables and figures. The only place that mention repetitions is a small secondary experiment in the appendix. If the main experiments were indeed repeated, the number of seeds should be specified, and CIs and error bars added. In case they were not, it is necessary to repeat them so that the results are reliable.

---

> ### Author Response · Authors · 2025-02-17
>
> **C1**: We appreciate your emphasis on statistical reliability. We have conducted experiments with three different random seeds for our method. Below are the test results (mean ± standard deviation) across all COPs under small/median/large budgets:
>
> | | TSP20         | TSP50         | TSP100        | CVRP20        | CVRP50        | CVRP100       | OP20            | OP50         | OP100        | KP50          | KP100         | KP200         |
> |---------------|---------------|---------------|---------------|---------------|---------------|---------------|-----------------|--------------|--------------|---------------|---------------|---------------|
> | 500           | 0.026 ± 0.007 | 0.267 ± 0.033 | 1.435 ± 0.179 | 0.390 ± 0.040 | 1.536 ± 0.092 | 2.817 ± 0.121 | -1.073 ± 0.041  | 0.462 ± 0.116 | 1.901 ± 0.279 | 0.036 ± 0.005 | 0.015 ± 0.001 | 0.017 ± 0.002 |
> | 1000          | 0.015 ± 0.002 | 0.191 ± 0.006 | 0.969 ± 0.088 | 0.337 ± 0.006 | 1.235 ± 0.053 | 2.273 ± 0.079 | -1.115 ± 0.026  | 0.180 ± 0.082 | 1.156 ± 0.177 | 0.028 ± 0.001 | 0.011 ± 0.001 | 0.013 ± 0.001 |
> | 2000          | 0.014 ± 0.001 | 0.170 ± 0.008 | 0.873 ± 0.042 | 0.323 ± 0.008 | 1.190 ± 0.048 | 2.180 ± 0.070 | -1.132 ± 0.008  | 0.110 ± 0.085 | 1.016 ± 0.153 | 0.025 ± 0.001 | 0.011 ± 0.001 | 0.011 ± 0.000 |
>
> The consistently small standard deviations across all problem sizes and types indicate that our method achieves stable performance across different random initializations, demonstrating the reproducibility of our method. Based on our experience and experimental observations with the POMO[1] backbone used in this work, it demonstrates good training and testing robustness. We suppose other baselines would show similar stability as our method.
> We commit to:
> 1. Add these detailed statistical results to the appendix
> 2. Include training curves with confidence intervals (similar to Figure 2)
> 3. Provide complete training statistics showing the stability of both loss and objective values over multiple runs.
>
> [1] Kwon Y D, Choo J, Kim B, et al. Pomo: Policy optimization with multiple optima for reinforcement learning[J]. Advances in Neural Information Processing Systems, 2020, 33: 21188-21198.
>
> **C2**: Thanks for your suggestion. We will add descriptions of MTL baselines to the appendix and clarify the key steps in complexity analysis in Table 1. Regarding why these MTL baselines underperform and our method works better, we identify several key factors:
> 1. Training COP neural solvers with RL inherently suffers from low sample efficiency and training efficiency. Even training a neural solver for a single problem size requires substantial samples and training epochs. When considering multiple types and scales of COPs, the challenge, sample requirements, and training time increase significantly.
> 2. The MTL baselines considered in the paper uniformly train all tasks together, leading to dramatically increased time per training step (as shown in GPU hours per epoch in Table 1). Under limited budgets, these methods complete fewer training epochs, resulting in infrequent model updates and inaccurate task relationship estimation. This insufficient training is particularly problematic for RL models.
> 3. Our method works because it only trains one task at a time, consuming minimal time per epoch. This enables extensive collection of task affinity data and frequent model updates, allowing thorough training through discovered positive transfer.
> Results in Table 1 and Table 2 support these points: MTL baselines with shorter per-epoch time (Naive-MTL, Bandit-MTL, UW) significantly outperform those with longer training times (PCGrad, Nash-MTL, and IMTL). Thank you again for pointing this out, we will include this analysis in the next version of the manuscript.
>
> **C3**: Our proposed method is not inherently limited to COPs. The theoretical analysis of our approach is independent of the problem domain, backbone architecture, and loss functions used. While our research focuses on learning to optimize and specifically on COPs, the method itself is more general. To demonstrate this generality, we conducted experiments in the time series domain, as reported in Appendix A.12. The results show varying effectiveness: our method achieves the best performance on 3 out of 7 datasets for Long-term Forecasting Tasks (Table 10).
>
> **C4**: We thank you for pointing out these typos. We will correct them in the next version of the manuscript.

---

> > ### Comment · Reviewer_W2Yt · 2025-02-17
> >
> > Thanks for your replies and clarifications. Regarding statistical validity, 10 seeds is generally considered a minimum standard. You may also be able to get away with fewer (e.g. 5) _if_ it is indeed the case that the methods lead to very stable performance. The key aspect is that in the final version you should have sufficient seeds so that confidence intervals of the methods don't overlap in most of the analyses. You could also take a look at https://proceedings.neurips.cc/paper_files/paper/2021/file/f514cec81cb148559cf475e7426eed5e-Paper.pdf.

---

> > > ### Author Response · Authors · 2025-02-17
> > >
> > > Thank you for your suggestion and the reference. Given the computational constraints (each method requires 6.64 days of GPU hours using 8 A100 GPUs), we hope you understand that we cannot update all results during the discussion period. However, we will promptly update the stability results of our method with at least 5 random seeds here, and we guarantee that the results for other baselines will be included in the final version of the manuscript to ensure reproducibility of our findings.

---

> > > > ### Comment · Reviewer_W2Yt · 2025-02-17
> > > >
> > > > Yes, just to be explicit, I do not expect this to be updated during the discussion period -- if the paper does end up securing acceptance, this can be part of the "final check".

---

> > > > > ### Comment · Reviewer_W2Yt · 2025-03-05
> > > > >
> > > > > Dear Authors,
> > > > >
> > > > > As the time to submit a decision recommendation is approaching, I wanted to check whether the results with 5 random seeds were added. I could not find them in the manuscript.

---

> > > > > > ### Author Response · Authors · 2025-03-06
> > > > > >
> > > > > > Dear Reviewer,
> > > > > >
> > > > > > Thank you for your reminder. We are currently running the experiments with 5 different random seeds as suggested. The results are expected to be completed in approximately three days. We will promptly update the manuscript with these additional results and notify you once it's done.

---

> > > > > > ### Author Response · Authors · 2025-03-19
> > > > > >
> > > > > > Dear Reviewer,
> > > > > >
> > > > > > We apologize for the delayed response due to technical issues with our equipment. We have updated Figure 2 in the latest version to illustrate the training stability of different methods across five repeated training runs.
> > > > > >
> > > > > > The results demonstrate that most MTL methods (with the exception of CAGrad) exhibit excellent training stability. Among these methods, our approach demonstrates superior performance with faster training convergence and better overall results.
> > > > > >
> > > > > > Thank you for your patience in awaiting these results. We hope that the delayed results will not adversely affect your decision regarding our work.
> > > > > >
> > > > > > Best,
> > > > > >
> > > > > > Authors

---

> ### Author Response · Authors · 2025-03-10
>
> Dear Reviewer,
>
> We have finished the reproduction experiments and updated the results in the revised version. The statistical results have been updated in Table 2 and Table 4, with confidence intervals for evaluation results on individual tasks presented in Figure 11 in Appendix A.12. Regarding the training stability analysis similar to Figure 2, unfortunately, our rented server unexpectedly crashed during the compilation process of the experimental results, resulting in the loss of these specific data.
>
> Nevertheless, we can infer some insights about training stability from the variance information for Average Gap in Table 2 and the evaluation error bars for individual tasks presented in Figure 11. These data suggest that certain methods demonstrate relatively higher training stability (STL, naive MTL, UW, and our proposed method), while others such as CAGrad and PCGrad appear to exhibit less stable training behavior (we have made the considered decision to remove the IMTL baseline from our comparison due to its highly unstable experimental results).
>
> We sincerely apologize for this unfortunate circumstance and hope you can understand the situation.  We are currently making every effort to recover these lost data, and should those efforts prove unsuccessful, we are fully committed to rerunning these baselines and providing comprehensive training stability results in a subsequent version of the manuscript.
>
> Best,
>
> The Authors

---

### Review · Reviewer_aEVN · 2025-02-14

**Summary Of Contributions:**

The paper presents a training paradigm for multi-task neural solvers addressing combinatorial optimization problems (COPs). The authors propose a loss decomposition framework under an encoder-decoder structure, which facilitates training via a MBA based task selection algorithm.

Empirically, the approach enhances training efficiency and generalization performance compared to traditional multi-task learning (MTL) methods and single-task learning (STL).

**Audience:**

Yes

**Claims And Evidence:**

No

**Requested Changes:**

1. Address the Cross-Task vs. Intra-Task Boost. Please correct me if I were wrong, I feel a bit confused as the empirical findings of overall performance improvements with the relatively low cross-task gradient overlap shown in the influence matrix. If the shared encoder or certain universal solution heuristics are playing a role, clarifying that mechanism would remove such counter-intuitive contradictions.

2. For certain tasks or scales, (see influence matrix) training them together in the same problem family (e.g., TSP-50 vs. TSP-100) may introduce conflicts. Could the authors bring some insights behind this?

3. Include (at Least Preliminary) Larger-Scale Experiments (e.g. TSP200 is enough for illustration). It would be interesting to check the generalization w.r.t. scale changes (say 100 to 200 in TSP problems). This would also strengthen claims of the method’s scalability and ability to generalize across a wide range of instance sizes.

4. Provide insight into why random sampling may outperform single-task schedules in some cases. If random sampling systematically explores all tasks more uniformly, it might balance knowledge accumulation in ways that a naive or a non-adaptive schedule does not. This could be an interesting angle to explore and clarify.

**Strengths And Weaknesses:**

Strengths:

1. Applying MAB to multi-task training for combinatorial optimization is interesting and relatively unexplored. It provides a principled mechanism for adaptively allocating training efforts among different tasks

2. The loss decomposition and the introduction of an influence matrix are conceptually appealing. They give a gradient-based explanation of how training one task impacts others (both within the same problem type and across different types).

Weaknesses:

1. (minor) While using MAB is promising, the motivation could be clearer. It would help if the authors provided more real-world scenarios or theoretical reasoning showing why an adaptive scheduler (bandit-based) is preferable to simpler heuristic schedules (e.g., uniform or priority-based sampling).

2. (major) The influence matrix shows that cross-task correlations (between different CO problem types) are fairly low, which seems to contradict the idea that training on multiple tasks can improve overall performance and generalization. This is a key question: If cross-task gradients are almost orthogonal, from where does the boost in performance come? A more thorough explanation or illustration of how partial shared knowledge (e.g., shared encoder parameters or certain patterns in solution construction) benefits all tasks would strengthen the paper.

3. (moderate) While the paper includes tasks up to size 100, testing on larger-scale instances (e.g., TSP-200, TSP-500) without training on them would provide stronger evidence of how the method scales in more realistic industrial settings. A pilot experiment could demonstrate whether the approach remains effective and whether the bandit selection shifts to favor smaller tasks or invests more in the large tasks over time.

---

> ### Author Response · Authors · 2025-02-17
> **Response Part 1/2**
>
> We sincerely thank the reviewer for the insightful comments and valuable suggestions. We will incorporate the followed discussions in the next version of our manuscript, particularly highlighting the analysis of influence matrix mentioned in RC1 & W2 and RC2, and the clarification of random policy performance in RC4. Below are our point-by-point responses.
>
> ### **RC1 & W2**:
> We appreciate this insightful observation. First, we want to emphasize that while the low correlation between different COP types may seem counter-intuitive, this is an **objective finding** under the given backbone (POMO) and training algorithm (RL) - the **first such observation** in the Learning to Optimize field. We believe this finding poses both challenges and opportunities for future neural COP solver design.
>
> Then we would like to clarify that the cross-problem type performance improvements do not stem from the shared encoder learning mutual improvement information. Rather, the improvements come from efficient training (see GPU hours/epoch in Table 1) and positive transfers in independent model parameter subspaces for each COP type: As observed through the influence matrix, different COP types correspond to nearly orthogonal subspaces in the model parameter space (discussed at the end of Section 4.3). Each subspace learns solution information for its corresponding COP type, though this allocates fewer training parameters to each COP type.
>
> Based on this understanding, we further enhanced our method in Appendix A.8 by grouping the four COP types for training (Ours-4G), achieving better results compared to training all 12 tasks together under the same experimental settings. However, it's important to note that **this grouping strategy is informed by the influence matrix obtained through our method - such grouping relationships are not known a priori in practical applications**.
>
> As is well known, MTL improves training efficiency and model performance by reducing negative transfer while enhancing positive transfer between tasks. Our results show minimal negative transfer between different COP types. As mentioned in our future work, this motivates the design of better backbones and training algorithms that can capture connections between different COP types to promote positive transfer.
>
> ### **RC2**:
> The scale generalization challenge for COPs has been noted in previous works, such as Figure 5 in [1] and Figure 3 in [2], which demonstrate that neural solvers trained on specific problem sizes show degraded performance on other scales, with performance declining as the gap between training and testing sizes increases. This fact is in turn verified (quantified) by the influence matrix in our work.
> This phenomenon indicates that for different scales of the same COP type, both positive and negative transfer exist simultaneously. The positive transfer enables a trained neural solver to maintain basic solving capabilities compared to random initialization. However, the negative transfer arising from scale differences hinders the neural solver's generalization ability across different problem sizes. This is also reflected in our Figure 4(b), where the visualization of influences throughout training reveals both positive and negative transfer between different scales of the same problem type.
>
> In our work, to maintain the generality of our method, we did not specifically address this negative transfer between different scales of the same problem type. However, we believe these observations provide valuable insights for future research on universal solvers for COPs.
>
> [1]. Kool W, Van Hoof H, Welling M. Attention, learn to solve routing problems![J]. arXiv preprint arXiv:1803.08475, 2018.
>
> [2]. Joshi C K, Cappart Q, Rousseau L M, et al. Learning the travelling salesperson problem requires rethinking generalization[J]. Constraints, 2022, 27(1): 70-98.
>
> ### **RC3 & W3**:
> We've investigated the model's scalability and generalization capabilities by testing on real-world datasets and larger instances, as detailed in Appendix A.9 and our response to Reviewer ZdYQ. Specifically,
> - TSPLib instances with sizes ranging from 52 to 442 nodes (Table 8 in A.9)
> - CVRPLib instances with sizes ranging from 23 to 181 nodes (Table 9 in A.9), and large-scale CVRP instances  ranging from 100 to 1000 nodes (response part 2/2 to Reviewer ZdYQ)
>
> These extensive experiments demonstrate our method's superior generalization capabilities across varying problem scales.

---

> > ### Author Response · Authors · 2025-02-17
> > **Response Part 2/2**
> >
> > ### **RC4 & W1**:
> > In Table 2, the random policy achieves competitive results compared to MTL baselines due to its efficient training strategy (sampling one task at a time). It slightly outperforms STL (in small and median budgets) because positive transfer exists between same-type COP during the early stages of model training. However, unlike our method which selects tasks that maximize positive transfers across all tasks, the random policy doesn't actively explore and exploit task relationships. This lack of strategic task selection explains why it underperforms compared to our approach.

---

### Comment · Action_Editor_KcW4 · 2025-01-16

Dear Authors,

Thank you very much for submitting your work to TMLR. Three reviewers have now been successfully recruited for this submission.

Given the conference deadlines in the coming weeks, some reviewers have asked for a two-week extensionto to complete their reviews. I believe this is a reasonable request, and therefore, the reviews are now expected to be completed by February 14.

I appreciate your understanding of this arrangement. Please feel free to reach out if you have any questions or concerns.

Best Regards,

AE

---

### Author Response · Authors · 2025-03-10
**Response to Promised Revisions**

Dear Reviewers,

Thank you for your valuable comments during the discussion period. We have implemented all the promised revisions and updated our manuscript accordingly. Below is a summary of the changes, with references to their corresponding locations in the main text:

> Low Cross-Task Correlations (RC1 & W2 for Reviewer aEVN, RC1 for Reviewer ZdYQ)

We have treated this observation as an objective finding rather than an issue with the proposed method. We believe this finding highlights both challenges and opportunities for future neural COP solver design. We have clarified this observation as an insight for future work at the end of Section 4.3.

> Clarifications on Performance Improvement (RC1 & W2 for Reviewer aEVN, RC2 for Reviewer W2Yt)

We have provided a detailed discussion of the performance improvements in Appendix A.9.

> Negative Training Effects within Same Type of COPs (RC2 for Reviewer aEVN)

We have addressed this point in Section 4.3, observation (1).

> Clarification on Random Policy (RC4 for Reviewer aEVN)

We have elaborated on this point at the end of the Results part in Section 4.1.

> Reproducibility Concerns (RC1 for Reviewer W2Yt)

We have repeated the training for all methods across 5 random initializations of model parameters. The statistical results have been updated in Table 2 and Table 4, with confidence intervals for evaluation results on individual tasks presented in Figure 11 in Appendix A.12.

> Details for MTL Baselines (RC2 for Reviewer W2Yt)

We have provided a comprehensive introduction to the MTL baselines in Appendix A.5.

> Experiments on Large-Scale CVRP Instances on CVRPLib X-set (RC5 for Reviewer ZdYQ)

The results have been included in Table 10 in Appendix A.10.

---

Best,

The Authors

---

### Decision · Action_Editor_KcW4 · 2025-03-18

**Recommendation:** Accept with minor revision

**Comment:**

The reviewers found the proposed framework/method interesting, generalizable, and conceptually appealing, with comprehensive and robust experimental analysis. While acknowledging these strengths, they raised many concerns and requested changes in their reviews. After rebuttal, most of the concerns raised were properly addressed, and all reviewers leaned toward accepting this work. In the official recommendation, reviewer W2Yt believes two revisions are still requested for the camera-ready version of this work: (1) the learning curves should be provided, and (2) the results with 2 standard deviations (~ 95% confidence interval) should be provided instead of 1 standard deviation.

I read the paper in detail and fully concur with the reviewers' consensus that this work clearly meets the TMLR acceptance criteria (solid and well-supported claims, potential audience). In addition, I also believe reviewer W2Yt's requested changes are reasonable. Therefore, I recommend accepting this work with minor revision.

I expect the authors can provide both the learning curves and experimental results with 2 standard deviation in the camera-ready version of this work.

**Audience:**

All reviewers believe some individuals in TMLR's audience could be interested in the findings of this paper.

**Claims And Evidence:**

This work proposes a general multi-armed bandit (MAB) based traning framework for learning a multi-task combinatorial neural solver. The main contributions are: (1) the efficient multi-task training framework, which is general and not specific for neural combinatorial optimization; (2) theoretical analysis of the proposed loss decomposition method and MAB-based training algorithm; (3) detailed empirical analysis for the neural solver by the influence matrix; and (4) experimental results demonstrating that the proposed method outperforms both single-task learning (STL) and multi-task learning (MTL) baselines across various problems.

All reviewers believe the claims made in this paper are supported by accurate, convincing and clear evidence.

---

> ### Author Response · Authors · 2025-03-21
>
> Dear AC and Reviewers,
>
> We sincerely thank the AC and all reviewers for their valuable feedback and for accepting our paper.
> We commit to implementing all requested revisions and will promptly submit the camera-ready version that addresses these requirements.
>
> Sincerely,
>
> The Authors